# AMA-Bench: Evaluating Long-Horizon Memory for Agentic Applications

**Yujie Zhao**[1] **Boqin Yuan**[1] **Junbo Huang**[1] **Haocheng Yuan**[1] **Zhongming Yu**[1] **Haozhou Xu**[1] **Lanxiang Hu**[1] **Abhilash Shankarampeta**[1] **Zimeng Huang**[1] **Wentao Ni**[1] **Yuandong Tian**[2] **Jishen Zhao**[1]

## Abstract

Large Language Models (LLMs) are increasingly used as autonomous agents in complex, long-horizon applications, where effective memory is critical for sustained performance. Yet existing memory benchmarks are largely dialogue-centric, while real agent memory consists of continuous agent-environment interaction trajectories composed of states, actions, observations, and tool outputs. To address this, we introduce **AMA-Bench** (**A**gent **M**emory with **A**ny length), a benchmark for evaluating long-horizon memory in realistic agentic settings. It combines (i) real-world agent trajectories from representative applications with expert-curated QA, and (ii) synthetic trajectories that scale to arbitrary horizons with rule-based QA. Our study shows that existing memory systems underperform because they fail to capture causal and objective information and rely heavily on lossy similarity-based retrieval. We further propose **AMA-Agent**, a memory system based on causality-graph construction and tool-augmented retrieval. AMA-Agent achieves $57.2\%$ accuracy on AMA-Bench, outperforming the strongest baseline by $11.2\%$. Resources are available at: https://ama-bench.github.io/.

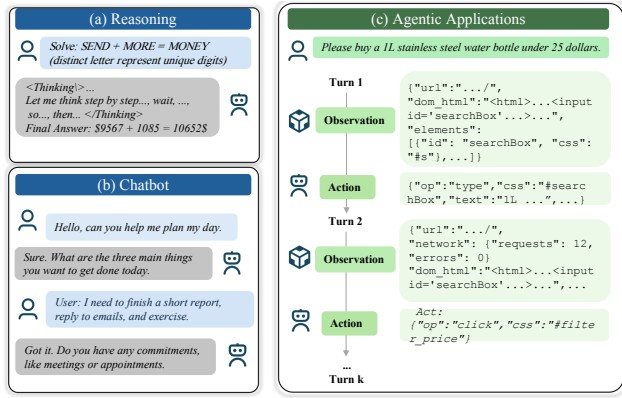

*Figure 1.* Comparison of memory across reasoning, chatbots, and agent applications. Agent trajectories exhibit unique properties, including being causally grounded, diverse symbolic artifacts, and dense objective information.

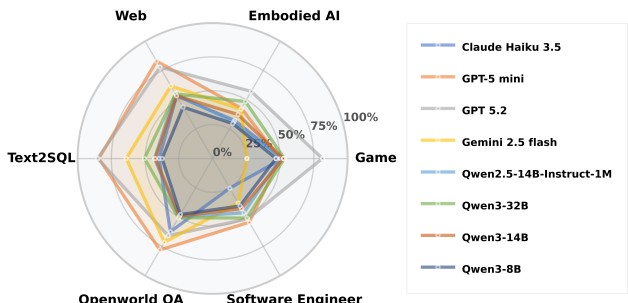

*Figure 2.* Model performance across agent task families in AMA-Bench.

## 1. Introduction

Large Language Models (LLMs) have rapidly evolved from solving closed-form reasoning tasks (Fig. 1 (a)) and serving as chatbots (Fig. 1 (b)), to serving as autonomous agents (Fig. 1(c)). Autonomous agents require long-horizon reasoning and experience reuse to complete tasks like open-space navigation, code editing, and web search. To empower LLMs with these capabilities, agent memory has become an im-

portant component of agent design to manage LLM context (Wang et al., 2025b; Agashe et al., 2024). Strong memory modules are expected to satisfy two core capabilities: (1) effective memory processing, where complete agentic trajectories are transformed into structured factual representations, such as summaries, fact tables, or embeddings (Edge et al., 2025; Packer et al., 2023; Liu et al., 2026); and (2) effective memory retrieval, reliably selecting and leveraging the most relevant memory to guide decision-making. Existing benchmarks typically evaluate these capabilities in dialogue-centric or synthetic retrieval tasks (Hsieh et al., 2024; Maharana et al., 2024), focusing on specific subcomponents, such as single- or multi-hop questions for memory retrieval or state updating and memory condensation questions for memory processing. There is a lack of benchmarks and eval-

[1]UCSD [2]Recursive. Correspondence to: Jishen Zhao <jzhao@ucsd.edu>. This work was conducted independently and is unrelated to Recursive's products, services, or commercial activities.

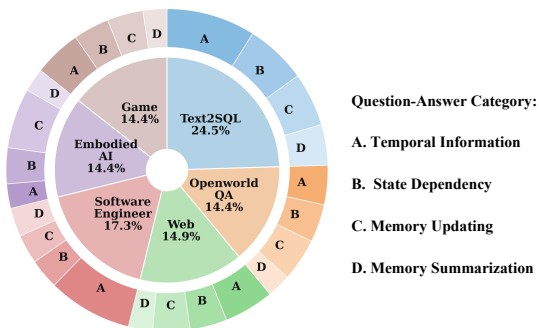

*Figure 3.* Domain and question type distribution in AMA-Bench.

uations of memory modules in long-horizon agentic tasks.

Real-world agents mainly operate in machine generated environments such as databases, code executors, and web interfaces, where they must process large volumes of *machine generated representations*. Yet, most existing memory benchmarks are still natural language centric and have three key limitations: (1) **A lack of representation types**: Agent trajectories encompass diverse machine generated symbolics (e.g., ASCII tables, JSON data, Unicode snippets, Python or HTML code blocks), whereas current benchmarks predominantly center on free-form natural languages; (2) **A lack of causality**: agent trajectories are causally grounded, where each action induces a latent environment state transition that constrains subsequent observations; but existing benchmarks follow unconstrained linguistic flow; (3) **Sparse objective information**: agent trajectories are machine-generated and objective, whereas dialog-centric benchmarks contain abundant redundant information such as phatic chit-chat.

To bridge this gap, we introduce AMA-Bench (Benchmarking Agent Memory with Any length), which comprises two complementary subsets: a real-world subset and a synthetic subset. The real-world component consists of expert-annotated and sanity-checked Question-Answer (QA) pairs sourced from six representative agent domains: Web, open-world QA, Text2SQL, Software Engineering, Gaming, and Embodied AI (see Fig. 3). Furthermore, we construct a synthetic subset in programmatic agent environments with automatically generated QA pairs. This design enables controlled synthesis of tasks at arbitrary horizons while keeping the agent-environment interaction pattern.

As shown in Fig. 2, our systematic evaluation indicates that agent memory remains challenging even for frontier commercial models, with GPT 5.2 achieving only 72.26% accuracy. Evaluating memory systems on AMA-Bench yields **three key insights**: (1) While existing agent memory techniques often outperform long-context LLM baselines on dialogue-centric benchmarks, they fall short to the baselines in many long-horizon agentic tasks, highlighting the unique diagnostic value of our benchmark (Sec. 4); (2) Our analysis reveals that suboptimal memory system design, rather than

base model capability, serves as the primary bottleneck for their poor performance (Sec. 4); (3) Existing lossy compression and similarity-based retrieval techniques are insufficient for the nuanced demands of agent memory, necessitating a paradigm shift toward agent-centric memory management strategies (Sec. 4).

Motivated by these insights, we present AMA-Agent, a framework designed to address the memory demands of agentic applications. Moving beyond lossy compression or similarity-based graphs, AMA-Agent implements a Causality Graph to preserve the objective information and explicit causal dependencies within interaction histories. To transcend the limitations of similarity-based retrieval, we introduce a Hybrid Tool-Augmented Retrieval mechanism. This approach enables more efficient information extraction and synthesis in machine-generated representations.

This paper makes the following main contributions:

**AMA-Bench.** We introduce the first benchmark suite built for evaluating memory in agent applications, AMA-Bench, with two complementary subsets: *Real world* preserves authentic machine-generated interaction patterns, and a *Synthetic* suite that enables controlled scaling of any horizon length and complexity.

**Comprehensive Evaluations.** Through comprehensive evaluation using AMA-Bench, we show that many existing agent memory designs underperform the long-context baseline, as errors introduced by lossy memory compression and similarity-based retrieval accumulate and compound over long-horizon agentic tasks; this highlights the critical need for agent-centric memory designs.

**AMA-Agent.** We propose AMA-Agent that addresses the identified bottlenecks with two mechanisms: (i) Causality Graph that preserves the integrity of objective information and causal dependencies, and (ii) Tool-Augmented Retrieval, which utilizes both graph node search and keyword-based search. Experimental results show that AMA-Agent outperforms the strongest existing memory baselines by 11.16% on average.

## 2. Related Work

### 2.1. Agent Memory Evaluation

We categorize existing memory benchmarks into two primary classes (as shown in Tab. 1): Dialogue-Centric and Long-Context. Dialogue-centric benchmarks evaluate memory retention over multi-turn human-agent interactions. LoCoMo (Maharana et al., 2024) and LongMemEval (Wu et al., 2025) evaluate long-term interactive memory in assistant-style chats; MemoryAgentBench (Hu et al., 2025b) tests multiple long-term memory competencies across diverse memory capabilities; MemoryBench (Ai et al., 2025)

*Table 1.* Comparison of memory benchmarks. NL: Natural Language.

| Category | Benchmark | Interaction Paradigm | Content Source | Average Length (tokens) | Representation Types | Memory Organization |
|---|---|---|---|---|---|---|
| Dialogue-Centric | LoCoMo (Maharana et al., 2024) | Dialogue | Real + Synthetic | 9K | NL + Vision | Episodic |
| | LongMemEval (Wu et al., 2025) | Dialogue | Synthetic | 115K | NL | Multi-session |
| | MemoryAgentBench (Hu et al., 2025b) | Dialogue | Real + Synthetic | 100K–500K | NL | Multi-domain |
| | MemoryBench (Ai et al., 2025) | Dialogue | Real + Synthetic | ~30–380K | NL | Multi-session + Feedback |
| | RealTalk (Lee et al., 2025) | Dialogue | Real | 17K | NL | Multi-day |
| Long-Context | QuALITY (Pang et al., 2022) | Long-Context | Real | 5K | NL | Multi-hop |
| | RULER (Hsieh et al., 2024) | Long-Context | Synthetic | 4K–128K | NL | Single-turn |
| | LongBench v2 (Bai et al., 2025) | Long-Context | Real | 8K–2M | NL | Document-level |
| Agent-Centric | **AMA-Bench** | **Agent-Env** | **Real + Synthetic** | **57K** | **NL + Machine** | **Trajectory-based** |

unifies diverse memory tasks into a continual learning suite; and RealTalk (Lee et al., 2025) grounds long-term memory evaluation in multi-day human dialogues. Long-context benchmarks such as QuALITY (Pang et al., 2022), RULER (Hsieh et al., 2024), and LongBench v2 (Bai et al., 2025) evaluate static document-level reasoning, focusing on multi-hop comprehension over long inputs rather than interactive or incremental memory. In contrast, AMA-Bench evaluates memory for agent applications, where the interaction trajectory is characterized by machine-generated representations, causal dependencies, and dense, objective information.

## 2.2. Agent Memory Mechanisms

Three main approaches have been explored to equip agents with long-horizon memory.

**Long-Context Models.** The first direction adapts LLMs to process memory directly as the context. For instance, GPT 5.2 (OpenAI, 2025) exposes an effective context window of approximately 400,000 tokens. On the open-source side, the Qwen2.5 1M (Yang et al., 2025b) series extends models to 100,000 tokens. Although simple and often strong in practice, this strategy remains bounded by physical context limits.

**Retrieval-Augmented Generation (RAG).** Another prominent research direction is RAG, which externalizes information into external storage during the memory construction stage and fetches relevant items based on similarity to augment the model's context during retrieval. Traditional methods, such as BM25 (Robertson & Zaragoza, 2009) and the Qwen3 Embedding series (Zhang et al., 2025b), store memory by partitioning data into discrete chunks. Structured RAG approaches have emerged to capture more complex relationships. For instance, GraphRAG (Edge et al., 2025) utilizes graph-based retrieval by constructing and aggregating entity-document graphs to capture these structural dependencies. Furthermore, HippoRAG2 (Gutiérrez et al., 2025) formalizes retrieval as a nonparametric form of Despite these advances, existing methods primarily rely on similarity-based or entity-centric retrieval, often neglecting the underlying causality within stored information.

**Memory Agent Systems.** Recent research has shifted from rule-based RAG pipelines to agent-centric memory management, where LLM agents autonomously decide how to perform memory construction and retrieval. MemoryBank (Zhong et al., 2023) enables models to autonomously summon and update the stored memories. MemGPT (Packer et al., 2023) formulates memory access as a decision problem, where the LLM learns when to retrieve and how to manage the long-term context. MemoRAG (Qian et al., 2025) proposes a dual system RAG architecture that maintains a global memory store and retrieves semantically similar clues to assemble a high-level draft for answering. MEM1 (Zhou et al., 2025), Mem-$\alpha$ (Wang et al., 2025a), Mem0 (Chhikara et al., 2025), MemAgent (Yu et al., 2025), and A MEM (Xu et al., 2025) construct memory by iterative compression or edit-style operations such as insertion, deletion, and modification, and then directly condition generation on the compressed memory. SimpleMem (Liu et al., 2026) introduced a structured compression pipeline that filters redundancy, organizes memories into hierarchies, and adaptively retrieves relevant contexts. However, memory compression and similarity-based retrieval perform poorly on agent memory tasks for two main reasons. First, most compression methods are designed for natural language, where redundancy and subjective fillers are common. However, agent trajectories contain dense, causally structured state transitions. Second, agent memories are often machine-generated representations, and similarity retrieval frequently fails to extract the required evidence.

## 3. AMA-Bench

In this section, we introduce AMA-Bench (Benchmarking **A**gent **M**emory with **A**ny length), a benchmark suite designed to evaluate memory systems in agent-centric memory. We first present a general problem formulation that abstracts agent-environment interaction and provides a unified definition of memory systems for agent applications in Sec. 3.1. Building on this formulation, Sec. 3.2 identifies which memory capabilities are essential for long-horizon decision-making and operationalizes them as evaluation dimensions. Finally, Sec. 3.3 describes how we construct AMA-Bench, including both real-world and synthetic subsets.

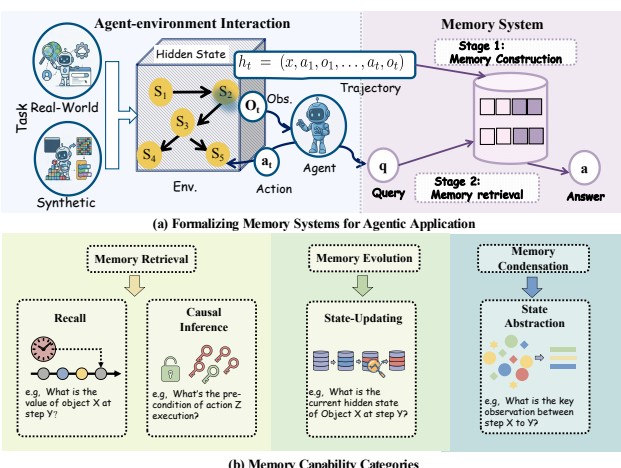

*Figure 4.* Formalizing memory system and capability for agentic applications.

## 3.1. Problem formulation

**Agent Environment Interactions.** We consider LLM agents operating within the reason and act paradigm (Yao et al., 2022; Shinn et al., 2023; Wang et al., 2023), where sequential decision-making is formulated as a Partially Observable Markov Decision Process (POMDP) defined by the tuple $\mathcal{M} = (\mathcal{S}, \mathcal{A}, \mathcal{O}, P, r)$ (See Fig. 4 (a)). At each time step $t$, the environment resides in a latent state $s_t \in \mathcal{S}$. Upon executing an action $a_t \in \mathcal{A}$, the agent receives an observation $o_t \in \mathcal{O}$ sampled from the observation function $O(s_t)$, and the environment transitions to $s_{t+1}$ according to the dynamics $P(s_t \mid s_t, a_{t+1})$. Given a task instruction $x$, the interaction generates a trajectory history $h_t = (x, a_1, o_1, ..., o_t)$. The partial observability motivates an explicit memory mechanism to persist the agent memory.

**The Memory System.** We formalized a memory system through two stages (see Fig. 4 (a)), memory construction (Build) and memory retrieval,(Retrieve). The construction stage, Build: $\mathcal{H} \rightarrow \mathcal{M}_{\mathrm{mem}}$, maps the interaction history $h_t$ to an external memory state $m_t \in \mathcal{M}_{\mathrm{mem}}$. The memory space $\mathcal{M}_{\mathrm{mem}}$ accommodates diverse structured representations, such as recursive summaries, knowledge graphs, and vector embeddings (Edge et al., 2025; Packer et al., 2023; Liu et al., 2026). Upon receiving a query $q_t$, the retrieval module extracts a query-relevant context $c_t = \mathrm{Retrieve}(m_t, q_t)$. The agent policy $\pi$ then determines the subsequent response based on the retrieved context and query: $a_t \sim \pi(\cdot \mid q_t, c_t)$.

## 3.2. Memory Capability Categories

The proposed formulation, supported by recent literature (Du et al., 2025; Zhang et al., 2024), underscores that an effective memory system must facilitate three core mechanisms: 1. Memory Retrieval: targeted access to the correct evidence. 2. Memory Evolution: continually updates memory as new observations arrive. 3. Memory Condensation: precisely

*Table 2.* Memory capability described in Sec 3.2. We group evaluation dimensions into three mechanisms and four capabilities.

| Mechanism | Capability | Description |
|---|---|---|
| **Memory Retrieval** | A. Recall | Identification of Temporal and sequential information. |
| | B. Causal Inference | Verification of action preconditions and dependency relations between states. |
| **Memory Evolution** | C. State Updating | Tracking updates to states, including explicit observations and hidden states. |
| **Memory Condensation** | D. State Abstraction | Filtering redundant content while extracting precise and condensed key information. |

extracting and condensing memory without information loss. Aligning these essential mechanisms with the specific requirements of agent-based tasks, we categorize memory capabilities into four functional capabilities. The formal definitions are detailed in Tab. 2, while illustrative examples are provided in Fig. 4 (b). These mechanisms encompass: Recall and Causal Inference (Retrieval), State Updating (Evolution), and State Abstraction (Condensation).

## 3.3. Benchmark Construction

With the above formulation and capability taxonomy, we now describe how we build AMA-Bench to jointly capture real-world complexity and provide controllable scaling complexity. AMA-Bench comprises two complementary components: (i) a real-world subset and (ii) a synthetic subset.

### 3.3.1. REAL-WORLD SUBSET

We curate high-quality, long-horizon trajectories from six representative real-world agent tasks, including web navigation, software engineering, text-to-SQL, embodied AI, gaming, and open-world tool with 2496 QA pairs (see Fig. 3 and Tab. 9 in Appendix A for the detailed category). For each task family, we gathered action-observation interaction traces from representative benchmarks using either state-of-the-art agent frameworks or expert-level trajectories provided directly by the environment. From this pool, we curated a subset for annotation, prioritizing longer trajectories while maintaining the original task distribution within each family. Specific details regarding the benchmarks and frameworks used are provided in Appendix A.

The real-world environments are treated as black boxes: we only observe agent-environment interaction logs (action and observation trajectories) and do not have access to the environment backend state. Building on the capability taxonomy in Sec. 3.2, we manually annotate each selected trajectory with 12 memory-intensive QA pairs that collectively cover all categories in Tab. 2. Each question is formulated such that its answer is supported by explicit and unambiguous evidence within the trajectory, ensuring that the correctness

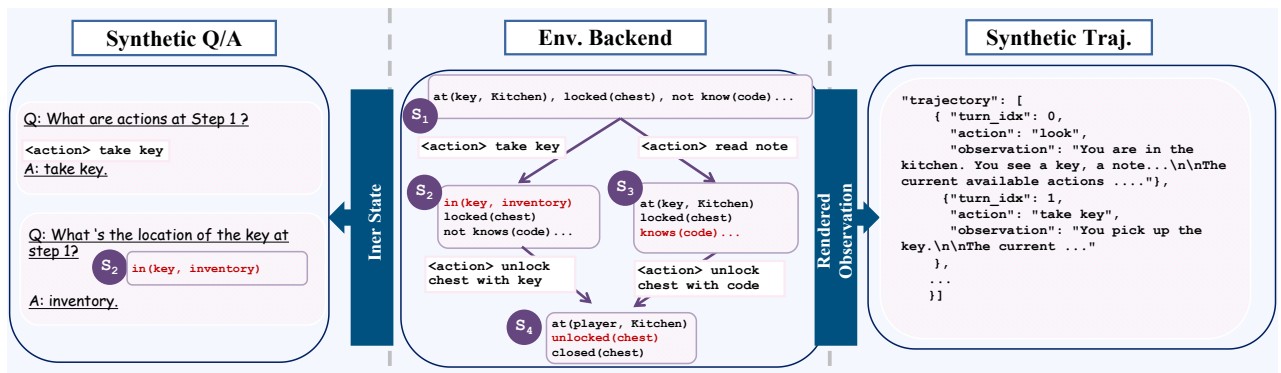

*Figure 5.* Synthetic subset construction pipeline. We synthesize an executable environment backend with explicit latent states and transitions, render machine-generated observations to form trajectories, and programmatically generate trajectory-grounded QA pairs.

of the question can be verified from the log itself.

QA pairs are then authored by graduate-level annotators with research experience in LLM agents, following shared guidelines that standardize evidence grounding and category coverage across six domains. To improve annotation reliability, each annotated trajectory undergoes a cross-review sanity check by a second annotator. This protocol yields expert-level QA annotations that are trajectory-grounded, category-aligned, and consistent across the task families. Examples of trajectories and QA are listed in Appendix L.1.

### 3.3.2. SYNTHETIC SUBSET

To systematically evaluate agent memory scaling, we construct a synthetic subset via programmatic environment synthesis. Each instance comprises an executable backend with controllable state transitions and a tunable perception interface, enabling the generation of verifiable trajectories with arbitrary horizons. Fig. 5 shows the pipeline of the synthetic subset construction. We synthesize tasks from two distinct environments characterized by long-range dependencies and partial observability: TextWorld (Côté et al., 2018), BabyAI (Chevalier-Boisvert et al., 2019). The description of the two tasks is listed in Appendix A.2.

**Environment Synthesis.** We parameterize each instance using a difficulty vector $\phi$ and a random seed to synthesize the environment backend. The latent state $s_t$ and transition kernel $s_{t+1} = P_\phi(s_t, a_{t+1})$ are programmatically defined and machine-verifiable. This allows us to systematically scale the interaction context length $L$ by increasing the environmental difficulty as dictated by $\phi$. For instance, in BabyAI, $\phi$ encompasses parameters such as the grid dimensions, the number of rooms, and the length of the instruction chain. By increasing the map size or adding nested dependencies, we can provably extend the trajectory length $L$.

**Trajectory Synthesis.** The synthetic nature of our environments grants full access to the environment MDP, enabling

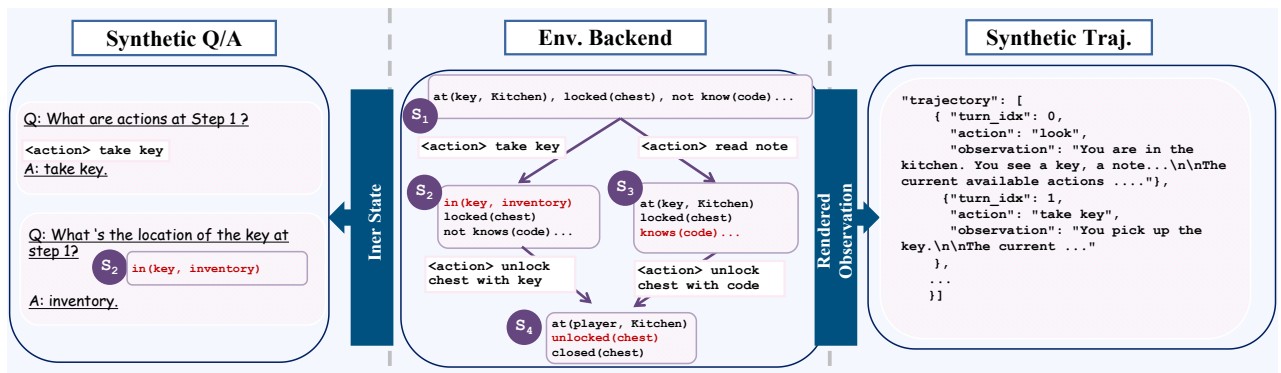

*Figure 6.* LLM-as-a-judge accuracy across different memory systems using Qwen3-32B as the base model.

the derivation of an optimal policy $\pi^*$ despite the agent's partial observability. We generate stepwise sequences $\{(a_t, o_t)\}_{t=1}^T$ grounded in these gold-standard transitions to resolve the issue of low-structural density. To further address representational diversity and robustness, we introduce two auxiliary perturbations beyond $\phi$: (1) Action Stochasticity ($\epsilon$): we inject random noise into $\pi^*$ to simulate sub-optimal action ratios, testing memory robustness under varying agent policies; and (2) Observation Verbosity ($\gamma$): we employ various symbolic representations with controllable descriptive granularity $\gamma$ for $o_t$.

**QA Synthesis.** Since we have access to the full MDP, we can programmatically generate golden QA pairs anchored to backend state variables such as state $s_t$ or transition kernel $s_{t+1} = P_\phi(s_t, a_t)$.

**Needle Synthesis.** Following the widely used needle-in-a-haystack (NIAH) paradigm (Nelson et al., 2024; Hsieh et al., 2024; Kamradt) for evaluating memory capabilities, we also instantiate a needle protocol in AMA-Bench. The *needle* here is the minimal set of trajectory turn IDs that contains all the evidence necessary to answer a query. Crucially, because AMA-Bench is backed by a programmatic environment, we can automatically synthesize and verify the needles. More details about the needle generation pipeline are listed in Appendix D.

# 4. Empirical Motivation

We benchmarked a broad set of representative memory systems on the AMA-Bench (see Fig. 6). The results reveal three key empirical insights that highlight the current limitations and directly motivate the design of our proposed method in Sec. 2.2.

**Motivation1:**
**Memory systems fall short of the long-context baseline.**
Fig. 6 compares representative memory systems against a long context baseline across six agent task families. A clear pattern emerges: the long context baseline is consistently strong and often achieves the best performance, whereas existing memory systems exhibit large variance across families and frequently underperform, even when they introduce structured memory construction or retrieval augmentation.

**Motivation2:**
**Memory Design bottlenecks the model performance.**

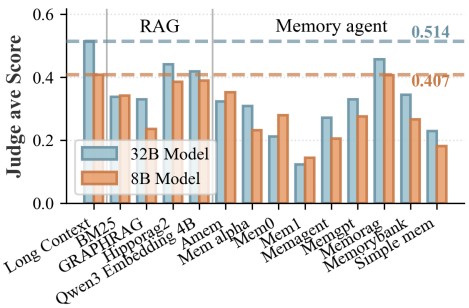

*Figure 7.* Impact of Model Scale vs. Memory Architecture. While scaling the backbone yields marginal gains, the choice of memory system accounts for the majority of performance variance.

A central question in building memory-augmented agents is whether performance bottlenecks reside in the backbone capacity or memory system design. Fig. 7 illustrates that scaling from 8B to 32B provides only marginal improvements (avg. improvement is 0.038), whereas varying the memory architecture induces significantly higher variance, with score ranges reaching 0.45.

**Motivation3:**
**Limitations of Existing Memory System Designs.**
To further pinpoint bottlenecks, we performed needle protocol ablation in BabyAI with three settings. *Full Observation (Needle)* provides the raw needle turns and serves as an upper bound. *Constructed Memory (Needle)* replaces them with method specific constructed memory to isolate construction loss. *End to End System* evaluates the full pipeline with retrieval.

Tab. 3 demonstrates two limitations. First, many methods degrade sharply after construction, e.g., MemoryBank drops

| Method | Full Observation w/ Needle ACC | Constructed Memory w/ Needle ACC | End to End System ACC |
|---|---|---|---|
| HippoRAG2 | | 0.37 (–19.6%) | 0.21 (–43.2%) |
| Mem1 | 0.46 | 0.29 (–37.0%) | 0.20 (–31.0%) |
| AMem | | 0.29 (–37.0%) | 0.24 (–17.2%) |
| MemoryBank | | 0.27 (–41.3%) | 0.26 (–3.7%) |

*Table 3.* Ablation study on performance bottlenecks under the needle retrieval protocol in BabyAI, evaluated by **accuracy (ACC)**. Dark red values denote relative decrease vs. the previous column.

by 41.3%, suggesting that compression tuned for redundant natural language fails to preserve dense state and causal information in agent memory. Second, similarity-based retrieval is unreliable: HippoRAG2 remains strong under constructed memory with needle turn but drops by 43.2% performance end-to-end.

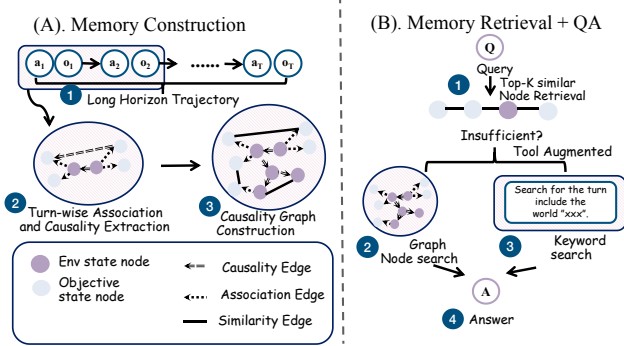

*Figure 8.* Overview of the AMA-agent. (A) Illustrates the transition from trajectories to a structured causality graph. (B) depicts the retrieval mechanism, utilizing tool-augment search.

# 5. The AMA-Agent

Motivated by the observations in Sec. 4, we develop the AMA-Agent with two core mechanisms: (A) a Causality Graph for memory construction to minimize information loss; and (B) a tool augmented retrieval module that complements standard retrieval with graph node traversal and keyword search to improve retrieval effectiveness.

## 5.1. Memory Construction: Causality Graph

AMA-Agent constructs a **Causality Graph** from the agent's trajectory. The construction proceeds in three stages: For each timestep $t$, the agent parses the adjacent turn pairs $(o_{t-1}, a_t, o_t)$ to extract environment and object states, identifying latent inter-state causal dependencies and state-object associations (Fig. 8 (A) ❶). These signals are instantiated as directed causality edges and undirected association edges connecting the respective state nodes (Fig. 8 (A) ❷). Finally, these local interactions are integrated into a global Causality Graph, where nodes are mapped into a latent embedding space to facilitate similarity-based retrieval and relational reasoning (Fig. 8 (A) ❸ ).

We emphasize that this extraction is implemented as an *LLM-based semantic abstraction* over the local trajectory segment $(o_{t-1}, a_t, o_t)$ (see prompt in App. K), rather than a rule-based parser over raw HTML DOM trees or program ASTs. The LLM is prompted to retain only *task-relevant* entities, state changes, and their latent dependencies; noisy low-level artifacts (e.g., layout markup or boilerplate stack traces) are intentionally discarded. For example, in a web task that asks for the bird species featured in a BBC Earth video, the extracted graph does not preserve raw page elements but only the higher-level entities "BBC Earth video", "penguins", the "search page", and the "generated report," along with the action edges that link them. Because each segment is abstracted independently, local extraction errors stay localized and do not cascade across the global graph; downstream retrieval can still recover the correct subgraph via either embedding similarity or the tool-augmented search path described next.

## 5.2. Memory Retrieval: Tool-Augmented Search

Beyond similarity-based retrieval, AMA-Agent adopts a tool-augmented search mechanism. It first retrieves the top $K$ nodes based on embedding similarity (Fig. 8 (B) ❶) and performs self-evaluation to assess whether the retrieved evidence is sufficient to answer the query. If the evidence is insufficient, the agent categorizes the missing context and invokes either the *graph node search tool* or the *keyword search tool*.

Under the *graph node search tool* route, the agent performs depth-controlled neighborhood traversal to aggregate multi-hop context and causal relations (Fig. 8 (B) ❷). Under the *keyword search tool* route, the AMA-Agent uses a tool interface that allows it to write and execute scripts for programmatic analysis, enabling precise keyword matching and statistical aggregation (Fig. 8 (B) ❸). Finally, the AMA-Agent synthesizes the retrieved evidence to produce a response (Fig. 8 (B) ❹).

In practice, tool invocation is selective rather than default. In an analysis over 200 randomly sampled QA pairs, the top-$K$ embedding retrieval was judged sufficient on its own in the majority of cases; the *graph node search tool* was invoked in only $10.0\%$ of cases (for multi-hop or precondition-dependent queries), and the *keyword search tool* in $23.5\%$ of cases (typically for trajectory-wide counting or pattern aggregation). This selectivity confirms that the two tools serve as complementary fallbacks rather than constant overhead.

# 6. Evaluation

## 6.1. Experimental Setup

**Benchmarks.** We evaluate our baselines on two complementary subsets: 1. Real-world Subset: This subset comprising a total of 2,496 QA pairs. 2. Synthetic Subset: we utilize two tasks with a total of 1,200 QA pairs. These tasks are stratified across five trajectory lengths (8K,16K,32K,64K, and 128K tokens), with 240 samples per interval.

**Baselines.** We consider three categories of baselines: long context models, RAG, and memory agents.

**Implementation Details.** To ensure a fair comparison, we evaluate all RAG baselines, memory-based agents, and our proposed AMA-Agent using the same backbone architectures: **Qwen3-32B** and **Qwen3-8B**. For each baseline, we adhere to the original authors' default embedding models and indexing configurations (refer to Appendix B for reproduction details). For AMA-Agent, we employ Qwen3-4B-embedding to map the causality graph into the latent space and set $K = 5$ for similarity-based node retrieval.

**Evaluation Protocol.** All experiments follow an *offline*, per-query evaluation protocol that is standard in prior memory benchmarks (Maharana et al., 2024; Hu et al., 2025b; Liu et al., 2026; Xu et al., 2025). For each trajectory, the memory system first ingests the *full* trajectory once to construct its memory state $m_T$. Each QA pair $(q_i, a_i^\star)$ is then answered independently against this fixed memory: $\hat{a}_i \sim \pi(\cdot \mid q_i, \text{Retrieve}(m_T, q_i))$. Earlier questions do not affect later ones, and the memory state is not updated across queries. This design isolates memory quality from execution dynamics and allows QA pairs to be evaluated scalably in parallel, in contrast to end-to-end agent rollouts that must run a strictly sequential interaction loop of $80 \sim 100$ steps per task.

**Metrics.** We report both Accuracy and F1-score on open-ended question answering. All QA pairs in AMA-Bench are formulated as open-ended generation rather than multiple choice, and Accuracy measures the fraction of predictions judged as correct by an LLM-as-judge based on Qwen3-32B. Additional details, cross-judge agreement (vs. GPT-5.2, GPT-5.4, Claude-4.6, and DeepSeek-v3.2), and human validation of the LLM-as-judge protocol are provided in Appendix C.

## 6.2. Key Results

**Real-world Subset.** We report the main results on the real-world subset in Tab. 4 (evaluating models with long contexts) and Tab. 5 (comparing different memory systems). While GPT 5.2 achieves the highest average accuracy (0.73) as Tab. 4 shows , its performance suggests that even strong commercial models have not fully mastered trajectory-based agent memory capabilities. Crucially, when controlled under the Qwen3 32B backbone (Tab. 5), **AMA-Agent** establishes a new state-of-the-art across all dimensions—Recall (0.6238), Causal Inference (0.6145), State Updating (0.5305), and State Abstraction (0.4719)—reaching an average of 0.5722. This significantly outperforms the strongest RAG baseline HippoRAG2 (0.4480) and the leading memory methods MemoRAG (0.4606) and EMem (0.4610) (Anonymous, 2025), the latter being a recent (Nov 2025) hierarchical mem-

*Table 4.* Performance of different models on real-world subset.

| Method | Recall | Causal Inference | State Updating | State Abstraction | Average |
|---|---|---|---|---|---|
| Claude Haiku 3.5 (Anthropic, 2025) | 0.4943 (0.3510) | 0.4507 (0.2792) | 0.4287 (0.3015) | 0.3090 (0.2648) | 0.4361 (0.3067) |
| GPT-5-mini (OpenAI, 2025) | 0.6951 (0.4010) | 0.7157 (0.3027) | **0.6575** (0.3288) | **0.6235** (0.3262) | 0.6784 (0.3464) |
| GPT 5.2 (Wailgum, 2025) | **0.7741** (0.4758) | **0.8047** (0.3512) | 0.6563 (0.3686) | 0.6037 (0.3582) | **0.7226** (0.3988) |
| Gemini 2.5 flash (Gemini Team, 2025) | 0.5834 (0.3682) | 0.5087 (0.2628) | 0.5000 (0.2395) | 0.4196 (0.2361) | 0.5168 (0.2878) |
| Qwen2.5-14B-1M (Yang et al., 2025b) | 0.5570 (0.4157) | 0.4111 (0.3209) | 0.4728 (0.3348) | 0.3368 (0.3560) | 0.4638 (0.3622) |
| Qwen3-32B (Yang et al., 2025a) | 0.6149 (0.4074) | 0.5178 (0.3289) | 0.4903 (0.3334) | 0.3657 (0.3172) | 0.5181 (0.3545) |
| Qwen3-14B (Yang et al., 2025a) | 0.5675 (0.3636) | 0.4430 (0.2931) | 0.4502 (0.3204) | 0.3176 (0.2716) | 0.4659 (0.3203) |
| Qwen3-8B (Yang et al., 2025a) | 0.5024 (0.3801) | 0.3776 (0.2830) | 0.3987 (0.3177) | 0.2923 (0.2792) | 0.4109 (0.3240) |

*Note: Results are reported as Accuracy (F1). The **best** and second best Accuracy values are highlighted.*

*Table 5.* Performance comparison of Agent Memory and RAG methods using base model Qwen-32B on real-world subset.

| Method | Recall | Causal Inference | State Updating | State Abstraction | Average |
|---|---|---|---|---|---|
| **RAG** | | | | | |
| BM25 | 0.3301 (0.1465) | 0.4264 (0.1549) | 0.3450 (0.1325) | 0.2498 (0.1623) | 0.3436 (0.1475) |
| Qwen3-Emb-4B (Zhang et al., 2025b) | 0.4843 (0.1590) | 0.4974 (0.1549) | 0.3520 (0.1353) | 0.3011 (0.1610) | 0.4227 (0.1522) |
| GraphRAG (Edge et al., 2025) | 0.3077 (0.2769) | 0.3905 (0.2634) | 0.3140 (0.2551) | 0.2879 (0.2588) | 0.3258 (0.2650) |
| HippoRAG2 (Gutiérrez et al., 2025) | 0.4579 (0.2356) | 0.5080 (0.1966) | 0.4403 (0.1892) | 0.3538 (0.1785) | 0.4480 (0.2048) |
| **Agent Memory Methods** | | | | | |
| MemAgent (Yu et al., 2025) | 0.2550 (0.1489) | 0.3380 (0.1606) | 0.2849 (0.1432) | 0.2202 (0.1655) | 0.2768 (0.1530) |
| Mem1 (Zhou et al., 2025) | 0.1180 (0.1857) | 0.1427 (0.1732) | 0.1205 (0.1659) | 0.1080 (0.2042) | 0.1229 (0.1807) |
| Amem (Xu et al., 2025) | 0.3084 (0.2707) | 0.3653 (0.2731) | 0.3088 (0.2480) | 0.2873 (0.2953) | 0.3186 (0.2695) |
| Mem0 (Chhikara et al., 2025) | 0.2011 (0.2413) | 0.2645 (0.2443) | 0.2101 (0.2225) | 0.1516 (0.2241) | 0.2104 (0.2343) |
| MemoRAG (Qian et al., 2025) | 0.4708 (0.1789) | 0.5497 (0.1811) | 0.4257 (0.1713) | 0.3659 (0.2073) | 0.4606 (0.1822) |
| MemGPT (Packer et al., 2023) | 0.3289 (0.1318) | 0.4404 (0.1475) | 0.2809 (0.1259) | 0.2526 (0.1431) | 0.3304 (0.1359) |
| Mem-alpha (Wang et al., 2025a) | 0.2876 (0.2325) | 0.4172 (0.1993) | 0.3064 (0.2000) | 0.2171 (0.2135) | 0.3117 (0.2130) |
| MemoryBank (Zhong et al., 2023) | 0.3231 (0.3128) | 0.4100 (0.2861) | 0.3006 (0.2678) | 0.3332 (0.3011) | 0.3397 (0.2928) |
| Simple mem (Liu et al., 2026) | 0.2012 (0.2039) | 0.1884 (0.1612) | 0.1764 (0.1594) | 0.1373 (0.1689) | 0.1811 (0.1764) |
| HiMem (Anonymous, 2026) | 0.2945 (0.2241) | 0.4018 (0.2065) | 0.3126 (0.1932) | 0.2238 (0.2087) | 0.2995 (0.2198) |
| EMem (Anonymous, 2025) | 0.4631 (0.2412) | 0.4925 (0.1878) | 0.4512 (0.1963) | 0.3421 (0.1824) | 0.4610 (0.2095) |
| **AMA-Agent** (AMA) | **0.6238** (0.3280) | **0.6145** (0.3103) | **0.5305** (0.2625) | **0.4719** (0.2825) | **0.5722** (0.2992) |

*Note: Results are reported as Accuracy (F1). The **best** and second best Accuracy values are highlighted.*

ory system. We further include HiMem (Anonymous, 2026) (Jan 2026), a near-deadline state-of-the-art baseline; AMA-Agent still outperforms it by over 27 absolute points on average. These results demonstrate that explicit modeling of long-horizon state dynamics and causal memory organization provides a more robust framework for agent reasoning than standalone retrieval-based or recent memory-agent approaches.

**Synthetic Subset**. We also evaluate all memory methods on the Synthetic subset and compare their scores against the real-world subset. Fig. 9 A illustrates the strong ranking correlation between real-world scenarios and our synthetic subset. The close alignment of most methods with the diagonal line demonstrates that the synthetic environment serves as a high-fidelity proxy for real-world performance, which is crucial given the high costs of real-world data acquisition and manual annotation.

Fig. 9 evaluates the performance stability of our method compared to other baselines as the sequence length increases from 8K to 128K. While the Long Context approach maintains competitive accuracy at shorter scales, its performance degrades significantly beyond 32K, revealing the inherent limitations of fixed context window. In contrast, AMA-agent exhibits superior scalability, maintaining robust and consistently high accuracy even at 128K.

### 6.3. Further Analyses

Due to space constraints, we defer to the appendix the per-trajectory efficiency comparison (App. G), per-difficulty TextWorld breakdown and qualitative behavioral case studies (App. H.1, H.3), out-of-benchmark generalization on LoCoMo (App. J), cross-model and cross-judge robustness (App. I, C.2), and the long-horizon difficulty statistics of

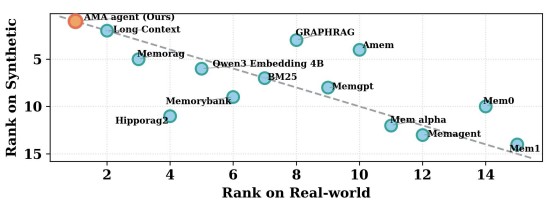

A. Correlation Analysis between Synthetic and Real-world Performance.

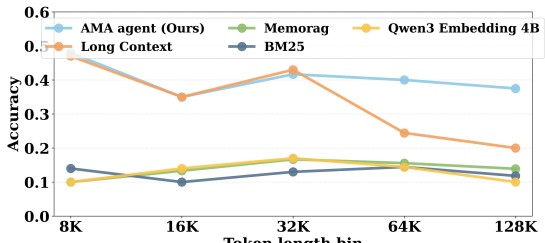

B. Scalability Analysis Across Trajectory Lengths

*Figure 9.* Performance Benchmarking. We evaluate 15 memory methods across Qwen 8B and 32B backbones.

*Table 6.* End-to-end task success and matched QA accuracy on TextWorld and Spider2, with the same Qwen3-32B execution agent and only the memory mechanism varied. Across both environments, QA accuracy and end-to-end success are strongly correlated, and AMA-Agent leads on every column.

| Method | TW E2E | TW QA | Sp2 E2E | Sp2 QA |
|---|---|---|---|---|
| **AMA-Agent** (AMA) | **51.5** | **40.4** | **26.2** | **57.4** |
| LongContext | 47.5 | 33.0 | 23.5 | 50.9 |
| HippoRAG (Gutiérrez et al., 2025) | 27.5 | 11.6 | 23.8 | 50.5 |
| Embedding (Qwen3-Emb-4B) (Zhang et al., 2025b) | 37.5 | 13.9 | 21.4 | 45.4 |
| MemoryBank (Zhong et al., 2023) | 31.5 | 11.9 | 17.2 | 26.0 |
| Mem0 (Chhikara et al., 2025) | 30.0 | 11.8 | 15.6 | 12.9 |

both subsets (App. A.1.1, A.2.1).

A natural concern with offline QA-based evaluation is whether QA accuracy on AMA-Bench actually predicts downstream agent success. To address this, we plug each memory system into the same vanilla Qwen3-32B execution agent on two interactive environments, the synthetic TextWorld and the real-world Spider2 (text-to-SQL), and report both end-to-end task success (**E2E**) and matched QA accuracy.

Tab. 6 shows that QA accuracy and end-to-end success are strongly correlated across methods on both TextWorld and Spider2: the Pearson correlation between the two reaches 0.96 on TextWorld and 0.98 on Spider2, computed over the six memory systems in the table. Three patterns emerge: (1) compression-based memory (MemoryBank, Mem0) is lossy for precise state tracking and collapses on both axes; (2) similarity-based retrieval (HippoRAG, Embedding) is fragile in the strongly causal TextWorld environment; (3) AMA-Agent gains because it preserves information fidelity through the causality graph while still supporting graph-based and tool-based retrieval.

*Table 7.* Ablation results for AMA-Agent

| Method | Recall | Causal Inference | State Updating | State Abstraction | Avg. |
|---|---|---|---|---|---|
| AMA-agent | 0.62 | 0.61 | 0.53 | 0.47 | 0.57 |
| w/o Causality Graph | 0.48 (−22.6%) | 0.48 (−21.3%) | 0.36 (−32.1%) | 0.35 (−25.5%) | 0.43 (−24.6%) |
| w/o Tool-Augmented Retrieval | 0.47 (−24.2%) | 0.51 (−16.4%) | 0.42 (−20.8%) | 0.31 (−34.0%) | 0.44 (−22.8%) |

*Note:* Values in dark red indicate the relative performance decrease compared to the full AMA agent.

### 6.4. Ablation Study

To validate the contributions of our key components, we perform ablation studies on the Causality Graph and tool augmented retrieval. The variant *w/o Causality Graph* replaces our structured Graph-based memory with a vanilla Qwen3 Embedding 4B indexes the context directly, while *w/o Tool Augmented Retrieval* disables tool calls and relies solely on embedding similarity retrieval. The results in Tab. 7 show that both modules are necessary for strong performance. Removing the Causality Graph causes a substantial degradation, with the average score dropping from 0.57 to 0.43, indicating that causality-aware representations are critical for agent memory. Likewise, removing tool augmented retrieval reduces performance to 0.44, suggesting that similarity search alone is insufficient and that tools provide complementary evidence access for robust reasoning.

### 7. Conclusion

In this paper, we introduced AMA-Bench to bridge the disparity between natural language-centric evaluations and the machine-generated, causally grounded nature of real-world agent trajectories. Our systematic analysis revealed that memory architecture is the primary determinant of performance, highlighting the limitation of lossy compression and similarity-based retrieval for dense, objective information. To address these challenges, we proposed AMA-Agent, which leverages a Causality Graph and Hybrid Tool Augmented Retrieval to significantly outperform sota baselines. A limitation of this study is its focus on in-episode memory; future work should extend these rigorous standards to cross-task scenarios involving lifelong learning.

## Impact Statement

This paper presents work whose goal is to advance the field of Machine Learning. There are many potential societal consequences of our work, none of which we feel must be specifically highlighted. Regarding the introduced benchmark, we confirmed that it was constructed entirely from open-source data sources. All data entries underwent human verification to ensure that no personally identifiable information (PII) or private data were included.

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

# Appendix

## A. Details of Dataset

Here, we provide a detailed introduction to the datasets used for evaluating the four core competencies, including the dataset curation, corresponding metrics, average context length, and a brief description.

### A.1. Real-world Subset

This section details the composition of the real-world subset, which comprises multi-turn trajectories curated from **six diverse domains** of agent-environment interactions (Tab. 8).

*Table 8.* Implementation details of collected agent trajectories across task families.

| Field | Benchmark | Trace Source | Total | Selected |
|---|---|---|---|---|
| **Embodied AI** | ALFWorld-verified (seen) | ALFRED (Li et al., 2025) | 140 | 33 |
| **Gaming** | BALROG / lmgame bench | BALROG / GamingAgent | 367 | 30 |
| **Web Task Execution** | WebArena (Zhou et al., 2023) | WebArena (Zhou et al., 2023) | 162 | 31 |
| **Software Engineering** | SWE-bench (Jimenez et al., 2023) | OpenHands (Wang et al., 2024) | 162 | 34 |
| **Open World Tool QA** | GAIA (Mialon et al., 2023) | CoSight (Zhang et al., 2025a) | 100 | 30 |
| **Text 2 SQL** | Spider 2.0 (Lei et al., 2024) | Spider2 agent (Lei et al., 2024) | 120 | 51 |

**Embodied AI.** We collect trajectories from both the **seen** and **unseen test splits** of ALFWorld (Shridhar et al., 2020b), a text-based embodied environment aligned with the ALFRED benchmark. These trajectories are generated using the expert-level demonstrations from ALFRED (Shridhar et al., 2020a) to ensure high-quality task completion in both familiar and novel environments.

**Gaming.** We curate gaming trajectories from two sources: BALROG (Paglieri et al., 2024), which includes Crafter (resource management), Baba is AI (long-horizon puzzle solving), and MiniHack (navigation); and LMGame-Bench (Hu et al., 2025a), which includes 2048 and Candy Crush. Trajectories are collected using GPT-5.1 with the BALROG agent framework and GamingAgent (lmgame-org, 2025) with memory and perception modules. For 2048, we use rule-based methods due to the extensive action sequences required. We select 30 trajectories totaling 360 QA pairs, with an average of 150 turns per episode.

**Web Task Execution.** We use WebArena (Zhou et al., 2023), a realistic web environment featuring fully functional websites across e-commerce, social forums, software development, and content management domains. Trajectories are collected using GPT-4.1 with the WebArena agent framework. We select 31 trajectories comprising 372 QA pairs, with an average of 25 turns and 34K tokens per trajectory, reaching up to 166K tokens for complex tasks.

**Software Engineering.** We collect trajectories from SWEBench Verified (Jimenez et al., 2023; OpenAI, 2024), which consists of real GitHub issues and pull requests from popular Python repositories. Trajectories are generated using Claude Sonnet 4 with the OpenHands framework (Wang et al., 2024), an open platform for AI software developers. We select 36 trajectories totaling 432 QA pairs, with an average of 103 turns and 19K tokens per trajectory.

**Open World Tool QA.** We use the GAIA benchmark (Mialon et al., 2023), which tests general AI assistants on real-world questions requiring reasoning, multi-modality handling, web browsing, and tool-use proficiency. Trajectories are collected using GPT-5 with the Co-Sight framework (Zhang et al., 2025a), which achieves state-of-the-art performance on open-sourced agent benchmarks. We select 30 trajectories across all three difficulty levels from the validation set, comprising 360 QA pairs with an average of 41 turns and 289K tokens – the longest among all domains, reaching up to 997K tokens.

**Text-to-SQL.** We collect trajectories from the Spider 2.0 benchmark (Lei et al., 2024), specifically sampling from the Spider2-Snow subset which focuses on enterprise-level text-to-SQL tasks with Snowflake databases. Spider 2.0 comprises three subsets (Snow, DBT, and Lite), with the Snow subset containing 547 examples. Among these, gold answers are provided for 120 examples to enable verification of generated SQL queries. We sample 51 trajectories from these verified examples to ensure answer correctness can be validated. Trajectories are generated using Claude Sonnet 4.5 with the Spider2-Agent framework, totaling 612 QA pairs with an average of 22 turns and 6K tokens per trajectory.

A comprehensive breakdown of the Real-world Subset is provided in Table 9. To provide a clearer illustration of our defined problem types, we present a representative example from the Web Task Execution domain (Figure L.1). This example shows how an agent must use different memory operations, such as tracking incremental UI changes, recognizing high level strategic failures, and handling long horizon interactions. Aligned with the three memory capabilities (Table 2), we define four QA categories that probe whether an agent has acquired the corresponding competencies required to answer them reliably, and we instantiate all four categories in this example. For additional qualitative visualizations of data samples, please refer to Appendix L.

*Table 9.* Statistics of QA pairs, evaluation type distribution, and interaction complexity.

| Field | #Samples | #QA | Evaluation Type | | | | Avg. Turns | Avg. Tokens | Max Tokens |
|---|---|---|---|---|---|---|---|---|---|
| | | | Type A | Type B | Type C | Type D | | | |
| Text 2 SQL | 51 | 612 | 223 | 153 | 134 | 102 | 21.80 | 6,049 | 10,718 |
| Open World Tool QA | 30 | 360 | 98 | 95 | 107 | 60 | 41.40 | 288,651 | 996,826 |
| Web Task Execution | 31 | 372 | 125 | 93 | 93 | 61 | 24.77 | 34,265 | 166,260 |
| Gaming | 30 | 360 | 120 | 90 | 90 | 60 | 149.87 | 14,909 | 33,360 |
| Embodied AI | 30 | 360 | 61 | 90 | 150 | 59 | 130.33 | 26,306 | 60,717 |
| Software Engineering | 36 | 432 | 212 | 75 | 73 | 72 | 103.22 | 19,296 | 28,615 |
| **TOTAL** | **208** | **2,496** | **839** | **596** | **647** | **414** | **73.29** | **57,506** | **996,826** |

### A.1.1. LONG-HORIZON REASONING STATISTICS OF THE REAL-WORLD SUBSET

To verify that the real-world subset truly stresses long-horizon memory rather than admitting trivial local answers, we sampled 720 QA pairs (120 per task, 28.8% of the full 2,496) and computed evidence-position and multi-hop statistics. Table 10 summarizes the result.

*Table 10.* Long-horizon reasoning statistics on a 720-QA sample of the real-world subset. Evidence positions are normalized to [0,1] over each trajectory.

| Metric | Value |
|---|---|
| Mean normalized evidence position | 0.475 |
| Mean hop count | 3.38 |
| Multi-hop ratio | 72.2% |
| Mean evidence span (turns, multi-hop only) | 16.1 |
| Mean normalized span (multi-hop only) | 0.260 |

*Table 11.* Position coverage of supporting evidence across the full real-world trajectory.

| Split | [0%,20%) | [20%,80%) | [80%,100%] |
|---|---|---|---|
| Overall | 21.4% | 62.9% | 15.7% |

The mean hop count of 3.38 and a 72.2% multi-hop ratio indicate that most real-world QA pairs require aggregating evidence from multiple non-adjacent turns. Furthermore, supporting evidence is broadly distributed across the trajectory (62.9% in the middle 60%) rather than concentrated at the beginning or end, consistent with the stress pattern targeted by AMA-Bench.

### A.2. Synthetic Subset

**BabyAI** are generated from the BabyAI environment (Chevalier-Boisvert et al., 2019), which supports six difficulty levels: `easy`, `medium`, `medium_hard`, `hard`, `very_hard`, and `hard_large`. Each trajectory is paired with 12 questions by default, and we collect a total of 50 trajectories, with an average length of 563 turns and 30,042 tokens per trajectory.

**TextWorld** are generated from the TextWorld (Côté et al., 2018) environment. We consider three game types: `coin_collector`, `cooking`, and `treasure_hunter`. The environment supports eight difficulty levels: `easy`, `medium`, `medium_hard`, `hard`, `very_hard`, `extreme`, `ultra`, and `mega`. On average, each trajectory contains 57 turns and 31,662 tokens.

To provide a concrete realization of the Synthetic subset described in Section 3.3, we present qualitative case studies from BabyAI and TextWorld. These examples are designed to illustrate how our programmatic framework evaluates specific agent capabilities by modulating synthesis parameters. For additional qualitative visualizations of data samples, please refer to Appendix L.

- **Probing Memory Robustness under Action Stochasticity ($\epsilon$):** Figure L.2 presents a diagnostic episode from the **BabyAI** environment under high action stochasticity ($\epsilon$). This setup evaluates the agent's ability to maintain the task goal within its interaction context when $\pi^*$ is perturbed by suboptimal exploratory noise. The observed failure—a task truncation—indicates that the agent's memory mechanism fails to distinguish gold-standard goal alignment from the increased "interaction noise," even when target objects are clearly rendered via the perception interface $O_\phi(s_t)$.

- **Evaluating State Tracking across Subgoal Chains ($\phi$):** Figure L.2 captures a failure in **TextWorld** that probes the agent's internal state-tracking of the backend transition kernel $P_\phi$. By scaling the difficulty vector $\phi$ to increase subgoal chain length, we test whether the agent can correctly update its "memory slots" based on transition events $\Delta s$. The repetitive invalid actions (e.g., attempting a `put` before a verified `take` event) reveal a breakdown in causal reasoning, where the agent loses track of the latent state $s_t$ despite having navigated the correct spatial transitions.

### A.2.1. SYNTHETIC EVIDENCE DISTRIBUTION

We additionally quantify whether the synthetic subset stresses long-horizon reasoning rather than admitting trivial local lookups. Table 12 reports the distribution of supporting evidence across the trajectory in 20% bins.

*Table 12.* Evidence distribution across normalized trajectory bins on the synthetic subset.

| Trajectory range | Turn % | Token % |
|---|---|---|
| [0%,20%) | 15.1% | 15.7% |
| [20%,40%) | 26.8% | 27.3% |
| [40%,60%) | 42.8% | 41.6% |
| [60%,80%) | 10.3% | 10.7% |
| [80%,100%] | 5.0% | 4.8% |

*Table 13.* Per-environment multi-hop statistics on the synthetic subset.

| Metric | Overall | BabyAI | TextWorld |
|---|---|---|---|
| Mean normalized evidence position | 0.424 | 0.476 | 0.420 |
| Mean hop count | 4.22 | 11.39 | 3.64 |
| Multi-hop ratio | 62.6% | 100.0% | 59.7% |

*Table 14.* Position coverage of supporting evidence across the full synthetic trajectory.

| Split | First 20% | Middle 60% | Last 20% |
|---|---|---|---|
| BabyAI | 9.3% | 89.4% | 1.3% |
| TextWorld | 15.2% | 79.7% | 5.0% |

The synthetic subset shows the same broad evidence distribution as the real-world subset (peak in the middle, not concentrated at trajectory boundaries), and a high multi-hop ratio in both environments (62.6% overall, 100% in BabyAI). Combined with the deterministic-backend QA generation pipeline (Sec. 3.3), this confirms that the synthetic subset captures meaningful long-horizon memory challenges rather than synthetic artifacts.

## B. Baseline Implementation Details

**Long-Context Model Baseline.** For long-context baselines, we directly *pack* the trajectory into the model input without retrieval or compression until reaching the maximum context length permitted by each API or checkpoint in Tab. 15. We

*Table 15.* Maximum context lengths used for long context baselines.

| Model | Max context tokens | Notes |
| --- | --- | --- |
| Claude 3.5 Haiku | 200,000 | API context window |
| OpenAI GPT 5 mini | 400,000 | API context window |
| OpenAI GPT 5.2 | 400,000 | API context window |
| Gemini 2.5 Flash | 1,048,576 | Max input tokens |
| Qwen2.5 14B Instruct 1M | 1,010,000 | Long context checkpoint |
| Qwen3 32B | 32,768 | Native |
| Qwen3 14B | 32,768 | Native |

reserve a fixed 4K-token budget for the model to generate the final answer, and use the remaining tokens as the effective input budget. When a trajectory exceeds this budget, we apply a simple truncation strategy that preserves both early and late interactions: we keep the first 50% and the last 50% budget length of the trajectory (by token count) and discard the middle portion to fit the context window.

### B.1. RAG Baseline

**GraphRAG.** constructs memory by using an LLM (Qwen3-8B/32B) to extract entities (object, location, action) and their relationships from trajectory text, storing them as a knowledge graph in parquet format. The trajectory is first chunked into semantic units of 15 turns per chunk with a maximum of 24,000 tokens. This construction process is inherently lossy, as it discards substantial raw trajectory details, particularly the detailed observation states and fine-grained action sequences present in the original data. During retrieval, GraphRAG selects the top-$k$ most relevant entities and relationships from the knowledge graph based on their descriptions, concatenating these structured elements into the prompt as context rather than including the full trajectory. We follow the default GraphRAG configuration with $k = 50$ entities and relationships, `max_gleanings=0`, and a description summarization disabled to preserve extraction fidelity.

**HippoRAG.** constructs memory by applying OpenIE-style extraction to trajectory text, yielding entities and relation triples that form a heterogeneous graph of passage (trajectory chunk), entity, and fact nodes; synonymy edges are added via nearest-neighbor search over entity embeddings. This construction is lossy because raw trajectories are distilled into triples and edges, potentially omitting fine-grained state transitions or action details. At retrieval time, HippoRAG computes query–fact similarity with dense embeddings, reranks top facts, maps them to linked entities, and runs personalized PageRank over the graph; the graph scores are combined with dense passage retrieval to select top-$k$ passages for QA. We use the default HippoRAG configuration with top-$k$ fact/entity linking $= 5$, passage retrieval top-$k = 200$, and QA context limited to the top 5 passages. All passage, entity, and fact embeddings use the same model BAAI/bge-m3.

### B.2. Memory Agents Baseline

**MemoryBank.** constructs a hierarchical memory by first chunking the trajectory into segments of 5,000 tokens with 500-token overlap, then using the LLM to summarize each chunk into a compact memory piece that preserves key subgoals, actions, observations, and failures. Each memory piece is embedded using a local sentence-transformer model (all-MiniLM-L6-v2), and a global summary is generated from all memory pieces to capture the overall strategy and critical facts. This summarization process is lossy, as fine-grained trajectory details are compressed into concise text. During retrieval, MemoryBank computes cosine similarity between the question embedding and memory piece embeddings, combined with an Ebbinghaus-inspired retention score that accounts for memory strength and recency of recall. The top-$k$ memory pieces are retrieved and concatenated with the global summary as context for answering. We use the default configuration with $k = 6$, forget decay $\tau = 5.0$, and strength increment of 1 upon each recall.

**MemAgent.** processes the trajectory as a stream of fixed-length sections, iteratively updating a recurrent memory state. For each chunk of 5,000 tokens, the LLM reads the current trajectory section along with the previous memory, then generates an updated memory that summarizes the agent's progress while retaining relevant details from earlier sections. This recurrent summarization is inherently lossy, as information from earlier chunks may be progressively compressed or forgotten as new sections are processed. During retrieval, MemAgent directly reads from its final accumulated memory without additional retrieval mechanisms, the memory itself serves as the complete context for answering questions. We follow the default configuration with a 4,096-token context window partitioned into: the current trajectory chunk (5,000 tokens, truncated if

needed), the accumulated memory (dynamically sized), and generation budget (1,024 tokens), with memory truncated from the beginning when context limits are exceeded to preserve more recent information.

**Mem-alpha.** employs a three-tier hierarchical memory architecture with an agentic approach to memory management. The system maintains: (1) Core Memory for high-level task understanding and rules, (2) Semantic Memory for storing factual knowledge as embedded vectors, and (3) Episodic Memory for recording specific events with temporal context. Trajectories are chunked using sentence-aware tokenization into segments of 4,096 tokens, preserving sentence boundaries. For each chunk, an agent equipped with memory tools (insert, update, delete, retrieve) autonomously decides which information to store and in which memory tier. Both semantic and episodic memories are embedded using text-embedding-3-small (1,536 dimensions) and retrieved via Top-K similarity search. During question answering, MemAlpha retrieves relevant memories using BM25 sparse retrieval, fetching the top-20 most relevant items per memory type. We follow the default configuration with a thinking budget of 1,024 tokens, maximum generation of 2,048 tokens, and memory consolidation occurring every 5 items.

**Mem1.** processes the trajectory through recurrent memory consolidation, iteratively updating a compact memory state with each new chunk of observations. For each chunk of 5,000 tokens, the LLM reads the current trajectory section along with the previous accumulated memory, then generates an updated memory that integrates new information while maintaining context and discarding redundant details. After processing all chunks, a final comprehensive summary is generated that consolidates key actions, important observations, overall progress, and patterns encountered. MEM1 directly uses this global consolidated memory as the complete context for answering all questions. We follow the default configuration with a 120,000-token maximum context window, trajectory chunks of 5,000 tokens, memory update budget of 1,024 tokens per chunk.

**Mem0.** constructs its memory layer through an LLM-driven fact extraction process, distilling raw trajectory data into a series of "atomic facts." These facts are subsequently embedded and stored in a vector database. To ensure memory consistency, Mem0 incorporates a conflict resolution mechanism that updates or replaces outdated information (e.g., evolving user locations). However, this extraction-based approach is inherently lossy for structured trajectory data, as it often omits critical low-level details. Empirical observations during our experiments indicate that bypassing the extraction layer and utilizing raw data directly can significantly enhance performance on trajectory-based benchmarks. During retrieval, Mem0 employs vector-based cosine similarity to identify the top-$k$ most relevant facts, which are then injected into the LLM prompt as context

**A-Mem.** implements a recurrent memory processing strategy to handle long-term trajectories. The input is segmented into chunks, and new memory states are built recursively by integrating the current chunk with preceding memory. Each memory entry consists of a concatenated representation of content, context, keywords, and tags, which is then embedded and stored in a vector database. This recursive construction, while thorough, introduces significant computational latency for long-context sequences. Furthermore, A-Mem supports memory evolution: it retrieves top-$k$ neighboring entries via vector search, and an LLM determines whether to establish new relational connections or update existing metadata. We set the `RECURRENT_CHUNK_SIZE` to 8,000 tokens.

**MemGPT.** implements a hierarchical memory architecture that separates memory into core memory (in-context) and archival memory (external storage with retrieval). The trajectory is inserted directly into archival memory as a complete text block, which is then indexed for retrieval. MemGPT uses an agentic approach where the LLM autonomously manages memory through function calls. It can search, insert, and retrieve from archival memory as needed during question answering. The archival memory is embedded using a local embedding model (BAAI/bge-small-en-v1.5) for vector-based retrieval. Unlike other memory agents that pre-process trajectories into summaries, MemGPT stores the raw trajectory text and relies on the agent's retrieval capabilities to fetch relevant portions at query time. During question answering, the agent receives the question and uses its memory tools to search archival storage, with retrieved content brought into the limited core memory context window. We use the default MemGPT configuration with auto-save disabled, maximum chaining steps set to 5 to prevent infinite tool-calling loops, and observations truncated to 8,000 characters when exceeding length limits.

**MemoRAG.** constructs memory by first building a global memory representation of the entire trajectory using a dedicated memory model, then enabling retrieval-augmented generation for question answering. The trajectory is converted to text format and processed by a memory encoder (Qwen2-7B-Instruct with beacon compression at ratio 4) that compresses the long context into a compact memory representation. This memory is then used to guide retrieval from the original text chunks. During retrieval, MemoRAG uses a dual-model architecture: the memory model generates retrieval cues based on the query and global memory, while a separate retriever (BAAI/bge-m3) fetches the top-$k$ most relevant chunks from the original

trajectory. The retrieved chunks are then passed to a generation model to produce the final answer. This approach is lossy during memory encoding, as the beacon compression mechanism reduces the original context to a fraction of its size. We use the default configuration with retrieval chunk size of 512 tokens, top-$k = 3$ retrieved hits, beacon ratio of 4, and maximum generation length of 256 tokens, retrieval via bge-m3.

**SimpleMem.** constructs memory through an LLM-driven extraction process that converts raw trajectory data into atomic memory entries. The trajectory is processed in sliding windows, where each window is passed to an LLM that extracts structured fields including lossless restatements with forced coreference resolution (eliminating pronouns and converting relative time to absolute timestamps), keywords, locations, persons, entities, and topics. These atomic entries are embedded and stored in a vector database. This extraction process is lossy, as trajectory details are abstracted into discrete semantic units. During retrieval, SimpleMem employs a hybrid strategy combining semantic vector similarity, lexical keyword matching, and symbolic metadata filtering. The system supports multi-query planning that decomposes complex questions into targeted sub-queries, and reflection-based refinement that iteratively checks information completeness and generates additional queries to fill gaps. We use the default configuration with parallel memory building (2 workers), parallel retrieval (3 workers), reflection enabled with maximum 2 rounds, and planning enabled for query decomposition.

## C. LLM-as-Judge Calibration Protocol

We include the following materials to ensure reproducibility and transparency of our LLM-as-judge evaluation.

### C.1. Judge Prompt and Output Format

We use QWEN3 32B as the primary evaluator. The judge receives the input triplet (question, reference answer, predicted answer) and returns a binary decision. The judge is required to output only one token in $\{$yes, no$\}$.

---

**Binary Correctness Judgement Prompt**

**System or Instruction Prompt**
You are an expert evaluator. You will be given a question, a reference answer, and a predicted answer.
Your task is to determine if the predicted answer is correct based on:
1. Factual correctness compared to the reference
2. Completeness of the answer
3. Relevance to the question
{context_str}

**Question:** {question}
**Reference Answer:** {golden_answer}
**Predicted Answer:** {predicted_answer}

Is the predicted answer correct? Respond with ONLY yes or no. Do not include any thinking process, explanation, or additional text.
**Answer**

---

### C.2. Human-Judge Agreement

To validate the reliability of our LLM-as-judge evaluation, we conducted human annotation on a sample of 300 instances (50 per subset) from the GPT-5.2 results. We obtain gold labels via independent human annotation. Each instance is labeled by at least two annotators with access to the question, reference answer, and predicted answer. Labels are binary: yes if the predicted answer is correct, otherwise no. Disagreements are resolved by majority vote, with a third annotator used for adjudication when needed.

Table 17 presents the confusion matrix and performance metrics aggregated across all subsets. The judge achieves 92.67% accuracy, indicating reliable alignment with human judgment.

We additionally cross-validate the Qwen3-32B judge against four heterogeneous LLM-as-judge alternatives on the same evaluation prompt. Table 16 reports pairwise agreement: Qwen3-32B agrees with all four within 8.1 points and aligns most closely with stronger commercial judges (GPT-5.4 at 92.8%, GPT-5.2 at 91.8%), giving us additional confidence that downstream method rankings are not sensitive to the choice of judge.

*Table 17.* Confusion matrix (left) and performance metrics (right) for LLM-as-judge vs. human annotations.

| | Human Label | | | | Metric | Value |
|---|---|---|---|---|---|---|
| **Judge Label** | Correct | Incorrect | Total | | Accuracy | 92.67% |
| Correct | 190 (TP) | 7 (FP) | 197 | | Precision | 96.45% |
| Incorrect | 15 (FN) | 88 (TN) | 103 | | Recall | 92.68% |
| | | | | | F1 Score | 94.53% |
| Total | 205 | 95 | 300 | | | |

*Table 16.* Pairwise agreement (%) of the primary Qwen3-32B judge against four alternative judges from different model families, measured on the same 300-instance human-validated sample.

| Judge pair | Agreement |
|---|---|
| Qwen3-32B vs. GPT-5.4 | 92.8% |
| Qwen3-32B vs. GPT-5.2 | 91.8% |
| Qwen3-32B vs. Claude-4.6 | 88.8% |
| Qwen3-32B vs. DeepSeek-v3.2 | 84.7% |

# D. Needle-in-a-Haystack QA Generation Pipeline

To evaluate the long-context retrieval capabilities of memory-augmented agents, we developed a structured pipeline to generate QA pairs where the answer is anchored to specific "needle" turns within a trajectory "haystack." The generation logic is formalized in Algorithm 1.

---

**Algorithm 1** QA Needle Generation for Trajectory Evaluation

---

**Require:** Source trajectory data $\mathcal{T}$, Bin sizes $\mathcal{B} \in \{8K, 16K, ..., 128K\}$, QA Types $\mathcal{Q} \in \{A, B, C, D\}$
**Ensure:** Final balanced dataset $\mathcal{D}_{final}$
1: $\mathcal{C} \leftarrow \emptyset$ {Initialize candidate pool}
2: $H \leftarrow \text{split\_by\_turns}(\mathcal{T})$ {Chunk haystack with unique turn identifiers}
3: **for all** $bin\_size \in \mathcal{B}$ **do**
4:     **for all** $qa\_type \in \mathcal{Q}$ **do**
5:         $\tau_{needle} \leftarrow \text{sample}(H, \text{strategy}=\text{"diversity\_first"})$ {Ensure depth diversity}
6:         $qa_{needle} \leftarrow \text{generate\_qa}(H, qa\_type, \tau_{needle}, bin\_size)$
7:         $qa_{needle}.\text{source\_ids} \leftarrow \{t.id \mid t \in \tau_{needle}\}$ {Map for traceability}
8:         **if** $\text{verify\_qa\_quality}(qa_{needle})$ **then**
9:             $\mathcal{C} \leftarrow \mathcal{C} \cup \{qa_{needle}\}$
10:         **end if**
11:     **end for**
12: **end for**
13: $\mathcal{D}_{final} \leftarrow \text{select\_balanced}(\mathcal{C}, \text{quota}=\{A:4, B:3, C:3, D:2\})$

---

**Key Strategies in Pipeline:**

- **Diversity-First Sampling:** Instead of random sampling, we pick "needle" turns from various depths of the trajectory (early, middle, and late stages) to prevent the LLM from exploiting positional biases.

- **Ground Truth Traceability:** By binding each QA pair to specific `turn_ids`, we can verify whether the agent's retrieval mechanism successfully identified the correct "needle" from the "haystack" during inference.

- **Balanced Distribution:** The final selection ensures that different reasoning types (e.g., spatial reasoning vs. object state tracking) are represented proportionally to avoid data skew.

# E. Example of needle turn for ablation study

To further illustrate the distinction between the three evaluation branches discussed in Section 5.3, we provide a concrete example from a Type A3 task (Object Visibility Change).

---

**Sample Query (Type A3)**

**Question:** Which object is no longer visible after the agent moves at step 1?
**Ground Truth:** After the agent executes the 'forward' action at step 1, 'a purple box' is no longer visible. It was visible at step 0, but the agent's movement changed its field of view.

---

- **Raw Observation Branch:**

```
[Turn 0] Action:  forward; Obs:  "In your view:  a yellow box, a purple box, a green
ball..."
[Turn 1] Action:  forward; Obs:  "In your view:  a yellow box, a green ball..."
```

*Note: The LLM must compare the two raw observation strings to infer the disappearance of the purple box.*

- **Oracle Memory Branch (and System Branch upon success):**

```
<memory>
- **Initial Position (Turn 0)**: Visible objects: Yellow box, purple box, green ball.
- **Progress (Turn 1)**: Action: Moved forward. Updated view: Yellow box and
  green ball visible; purple box no longer in sight.
- **Inference**: The disappearance of the purple box suggests movement progress.
</memory>
```

*Note: The state change is explicitly summarized. In the Oracle branch, this shard is force-fed to the LLM; in the System branch, the agent must retrieve this specific shard from the database.*

# F. Case-study

To further illustrate the distinction between our raw data storage approach and the default Mem0 extraction process discussed in Section 3, we provide a comparative analysis using a $Qwen3-32B$ model. This example demonstrates why LLM-driven "fact distillation" is inherently lossy for structured agent experiences.

---

**Case Study 1: Narrative Data (Successful Extraction)**

**Input:** My name is John. I like pizza and I work as a software engineer. I have a dog named Max.
**Status:** Success (4 Facts Extracted)

- `Fact 1`: Name is John
- `Fact 2`: Likes pizza
- `Fact 3`: Works as a software engineer
- `Fact 4`: Has a dog named Max

---

**Case Study 2: Trajectory Data (Extraction Failure)**

**Input (Abridged Trajectory):**

```
[Turn 0] Action:  look; Obs:  "You are in the middle of a room.  Looking
quickly around you, you see nothing.  Available actions:  go to cabinet 1, go
to cabinet 2..."
[Turn 1] Action:  go to toilet 1; Obs:  "You arrive at toilet 1.  On the toilet
1, you see a soapbottle 2..."
```

**Status:** Failure (0 Facts Extracted)

---

- **Narrative Extraction Logic:** Mem0's extraction prompt is optimized for high-level entities and static properties. In Case 1, the LLM successfully maps the input into an atomic "subject-predicate-object" structure, which is ideal for standard user profiling.

- **The Bottleneck in Trajectory Data:**

```
<internal_log>
– Input length: 5022 characters (15 turns)
– Model: Qwen3-32B
– Observation: The extractor fails to identify "facts" within the
  dynamic action-state-observation loops. Critical spatial
  identifiers (e.g., "toilet 1") are ignored as low-level noise.
</internal_log>
```

*Note: As evidenced by Case 2, the extraction-based approach is inherently lossy for agent-based tasks. By bypassing the* infer=True *extraction layer and utilizing raw trajectory segments directly in our* **System Branch**, *we preserve the environmental context that is otherwise discarded by the LLM-driven fact extractor.*

## G. Efficiency Analysis

A practical concern with structured memory is whether the additional graph-construction step makes the system uncompetitive in latency. Table 18 compares per-trajectory wall-clock cost on the real-world subset against six representative baselines, all run under the same Qwen3-32B backbone.

*Table 18.* Average per-trajectory latency (seconds) on the real-world subset, broken down into memory construction and retrieve+answer. AMA-Agent is more accurate than every faster baseline and faster than every more-accurate baseline.

| Method | Avg. Acc | Construct | Retrieve+Ans | E2E |
|---|---|---|---|---|
| Mem0 (Chhikara et al., 2025) | 0.2104 | **1.12** | 1.73 | **2.85** |
| MemAgent (Yu et al., 2025) | 0.2768 | 4.86 | **0.68** | 5.54 |
| **AMA-Agent** (AMA) | **0.5722** | 6.80 | 1.70 | 8.50 |
| HippoRAG2 (Gutiérrez et al., 2025) | 0.4480 | 10.90 | 1.20 | 12.10 |
| Mem1 (Zhou et al., 2025) | 0.1229 | 21.57 | 0.46 | 22.03 |
| SimpleMem (Liu et al., 2026) | 0.1811 | 23.55 | 0.80 | 24.35 |

Two observations: (1) AMA-Agent introduces moderate construction overhead relative to Mem0/MemAgent, but the cost is more than offset by a 2–5× accuracy gain. (2) Compared with stronger memory systems (HippoRAG2, Mem1, SimpleMem), AMA-Agent is simultaneously *more accurate* and *faster end-to-end*, occupying a strictly better point on the latency–accuracy frontier. A complementary QA-vs-rollout latency comparison is provided in Appendix H.2, where end-to-end rollouts can be $\sim 10^3 \times$ slower per task because of the sequential interaction loop – further motivating QA-based evaluation as the primary protocol of AMA-Bench.

## H. Additional End-to-End Analyses

This appendix expands the end-to-end experiments summarized in Sec. 6.3. We report (i) a per-difficulty breakdown on TextWorld, (ii) a QA-vs-rollout latency comparison that motivates QA-based evaluation as the primary protocol, and (iii) qualitative case studies that illustrate *why* causality-graph memory translates into better agent behavior.

### H.1. TextWorld End-to-End Success by Difficulty

We further stratify the TextWorld end-to-end results from Tab. 6 across the four canonical difficulty tiers. Table 19 shows that AMA-Agent degrades the most gracefully as horizon and causal-chain length grow.

*Table 21.* Qualitative comparison of memory categories on TextWorld behavior. RAG retrieves semantically relevant but causally incomplete evidence; agentic-summary compresses away preconditions; causality-graph preserves prerequisite chains and recovers from errors.

| Behavior pattern | RAG-based memory | Agentic-summary memory | Causality-graph memory (Ours) |
|---|---|---|---|
| Reduce repetitive actions | Retrieves semantically related clues such as "antidote" or "alchemy room" but misses the prerequisite chain (retrieve key → unlock basement → enter room), leading to repeated invalid `open door` attempts. | Compresses the trajectory into a coarse goal ("craft antidote in basement"), omitting executable preconditions; the agent repeatedly retries futile actions. | Preserves action dependencies explicitly, so the agent first retrieves the key, then unlocks the basement, then crafts the antidote, with no wasted attempts. |
| Recover from earlier mistakes | Retrieval still surfaces stale-but-relevant fragments, causing the agent to keep following an already-invalidated plan. | A summary may preserve the original plan but fail to reflect an irreversible state change (e.g., a consumed item). | Updates the consequence chain on each transition, so the agent abandons the invalid branch and re-plans. |

*Table 19.* TextWorld end-to-end task success (%) by difficulty, same Qwen3-32B execution agent with only the memory mechanism varied.

| Method | Easy | Medium | Hard | Very hard | E2E |
|---|---|---|---|---|---|
| **AMA-Agent** (AMA) | 84 | 61 | **35** | **22** | **50.5** |
| LongContext | **85** | 55 | 32 | 18 | 47.5 |
| HippoRAG | 63 | 30 | 12 | 5 | 27.5 |
| Embedding | 68 | 48 | 26 | 8 | 37.5 |
| MemoryBank | 53 | 36 | 23 | 14 | 31.5 |
| Mem0 | 50 | 35 | 20 | 12 | 30.0 |

LongContext is competitive on Easy tasks but its advantage collapses as difficulty grows, while AMA-Agent's gap over the next-best method widens from $-1$ point on Easy to $+3$ and $+4$ points on Hard and Very-hard, indicating that the causality graph becomes more valuable as causal chains lengthen.

## H.2. QA-vs-Rollout Latency

A central design choice in AMA-Bench is to evaluate memory through QA over a pre-collected trajectory, rather than full end-to-end rollouts. Table 20 measures the cost of both protocols on TextWorld and shows why this choice is necessary at the scale of AMA-Bench.

*Table 20.* Average and maximum wall-clock latency (seconds) per QA vs. per end-to-end task on TextWorld. End-to-end rollouts are $\sim 10^3 \times$ slower per task because the agent must execute 80–100 sequential interaction steps with the environment.

| Method | Avg / QA | Avg / task | Max / QA | Max / task |
|---|---|---|---|---|
| LongContext | 1.4 | 2,176 | 2.2 | 2,932 |
| HippoRAG | 12.1 | 3,660 | 47.2 | 7,290 |

Beyond raw cost, QA evaluation also offers cleaner failure attribution: a wrong answer is directly traceable to memory quality, whereas a failed rollout entangles memory errors with policy and tool-use errors. This is why AMA-Bench primary protocol is QA-based, with end-to-end results (Tab. 6) serving as a complementary downstream check.

## H.3. Qualitative TextWorld Behavior Case Studies

To illustrate *how* memory design changes agent behavior, we contrast three memory categories on representative TextWorld episodes. Table 21 summarizes typical failure modes for RAG-based memory, agentic-summary memory, and the causality-graph memory used by AMA-Agent.

Overall, RAG-based memory tends to fail by retrieving *semantically* relevant but *causally* incomplete evidence, while compression-based memory tends to fail by discarding preconditions or irreversible state changes. The causality graph addresses both by preserving prerequisite edges and updating state nodes online.

## I. Cross-Model Robustness

To rule out evaluation bias from sharing the same Qwen3-32B model for both the agent and the LLM-as-judge, we additionally evaluate every method with a different agent backbone – (i) the smaller Qwen3-8B on the real-world subset, and (ii) GPT-5-mini, from a different model family, on the synthetic subset. Table 22 shows the relative ordering of methods is preserved and AMA-Agent remains strongest in all three settings.

*Table 22.* Average accuracy across base models. Real-world numbers use the full real-world subset; the GPT-5-mini column uses the synthetic subset. The relative ordering among methods is preserved across all three settings.

| Method | Qwen3-32B | Qwen3-8B | GPT-5-mini (syn.) |
|---|---|---|---|
| Mem0 | 0.2095 | 0.2770 | 0.1141 |
| HippoRAG2 | 0.4378 | 0.3836 | 0.0732 |
| Mem1 | 0.1218 | 0.1429 | 0.0928 |
| MemoryBank | 0.3417 | 0.2648 | 0.0725 |
| A-mem | 0.3208 | 0.3498 | 0.1158 |
| **AMA-Agent** (AMA) | **0.5629** | **0.4814** | **0.2075** |

We note that synthetic-subset scores in the GPT-5-mini column are uniformly lower because the synthetic subset uses longer horizons and machine-rendered observations that are harder for any single-call agent backbone; the gap between AMA-Agent and the next-best method, however, *widens* in this harder regime, further supporting the value of structured causality-aware memory.

## J. Out-of-Benchmark Generalization on LoCoMo

Beyond the agent-trajectory setting of AMA-Bench, we additionally evaluate AMA-Agent on the established LoCoMo (Maharana et al., 2024) dialogue-memory benchmark using the same Qwen3-32B backbone to verify that its gains generalize beyond agent trajectories. Although LoCoMo is dialogue-centric and structurally different from agent-trajectory data, AMA-Agent still wins, indicating that the causality-graph plus tool-augmented design captures a more general memory principle rather than overfitting to AMA-Bench.

*Table 23.* Out-of-benchmark generalization on LoCoMo (Maharana et al., 2024) (Qwen3-32B). AMA-Agent outperforms the strongest baseline HippoRAG by 4.1 absolute points.

| Method | LoCoMo Acc. |
|---|---|
| A-mem (Xu et al., 2025) | 60.84 |
| Mem0 (Chhikara et al., 2025) | 63.51 |
| GraphRAG (Edge et al., 2025) | 65.39 |
| HippoRAG (Gutiérrez et al., 2025) | 72.10 |
| **AMA-Agent** (AMA) | **76.20** |

## K. Prompt Templates for AMA-Agents

This appendix provides the prompt templates used by AMA-agent. The construction phase compresses the trajectory into a structured state memory (COMPRESS_PROMPT_TEMPLATE). The retrieval phase performs chunk sufficiency judgement (CHUNK_SUFFICIENCY_JUDGMENT_PROMPT_TEMPLATE) and, when needed, trajectory-based code generation (CODE_GENERATION_PROMPT_TEMPLATE).

## Memory Construction Prompt Template

```
COMPRESS_PROMPT_TEMPLATE = """You are presented with a section of agent
trajectory (actions and observations). Compress it into a state memory that
future readers can use to answer detailed questions about what happened.

Task: {task}

Trajectory Section:
{trajectory_text}

{previous_state_text}

Identify KEY turns -- turns where state meaningfully changed, a
command/query/code was executed, an important result/value/schema/error
appeared, or the agent shifted sub-goal. For each key turn record an
env_state block with bullet keys that BEST FIT this turn. Common keys
include: action, command, query, location, inventory, schema, sql, result,
finding, error, change. Use whatever keys fit; do NOT force fields that do
not apply, and do NOT dump unrelated context into a "change" field.

CRITICAL -- copy these VERBATIM from the trajectory; never paraphrase or
summarize:
- commands, queries, code (SQL, shell, Python, function/tool calls)
- identifiers: file paths, URLs, table/column/schema names, IDs, UI element
  ids and selectors, entity names
- numeric values, counts, computed results, dates, response codes
- error messages

Also write a one-line Memory Summary describing the section's overall
progress.

Output everything after the marker below.

**STATE_MEMORY**

memory_summary: key_turns=<turn ids>; key_objective=<main objective>;
key_events=<critical events, commands, state changes, key results>

turn_id: <turn number>
env_state:
- <key>: <value>
- ...

turn_id: <turn number>
env_state:
- <key>: <value>

"""
```

## Chunk Sufficiency Judgement Prompt Template

```
CHUNK_SUFFICIENCY_JUDGMENT_PROMPT_TEMPLATE = """You are routing a question
about an agent trajectory to the right retrieval stage.

Query: {query}

Retrieved Turns (top-k from similarity search -- this is NOT the full
trajectory):
{retrieved_chunks}

IMPORTANT: the retrieved turns above are a small subset of the full
trajectory selected by similarity. Counts, lists, and aggregations computed
over this subset WILL UNDERCOUNT and MUST NOT be used to answer "how many"
/ "count" / "list all" type questions.

# Decision procedure
Step 1. Does the question ask any of the following?
  - "how many ...", "count of ...", "total number of ...", "frequency of ..."
  - "how often", "how many times did X happen"
  - "list all X", "find every X", "all distinct/unique X"
  - sum / average / percentage / aggregation
  - patterns spanning many turns ("redundant loop", "repeated actions",
    "every time X")
  - a tally or breakdown of tool/command/action types across turns
  -> If YES, you MUST choose NEED_CODE. Do NOT answer from the retrieved
     subset even if it appears to contain examples -- you will undercount.
     End here.

Step 2. Otherwise, can you answer the question COMPLETELY and ACCURATELY
using only the retrieved turns above (plus general knowledge)?
  - "Completely": every part of the question is addressed.
  - "Accurately": you can point to specific turn(s) in the retrieved set
    that justify each part of the answer.
  -> If YES, choose SUFFICIENT and provide the full answer. End here.

Step 3. Are there specific other turns (adjacent, range, or by index) you
need to inspect to answer? For example, the answer hinges on a turn that is
referenced but not shown, or you need context around a retrieved turn.
  -> If YES, choose NEED_GRAPH and specify which turns. End here.

Step 4. If you reach this step, the question requires reasoning over many
parts of the trajectory that the similarity search did not surface.
  -> Choose NEED_CODE.

# Output formats -- use EXACTLY one of these, on its own line(s) at the
# start of your response

SUFFICIENT
ANSWER: <your complete and accurate answer>

NEED_GRAPH: <spec>
  where <spec> is one or more of:
    turn_5 before=2 after=1
    turn_8 before=3 after=0, turn_15 before=0 after=2
    turns 5 to 10
    turns 3 to 8, turns 15 to 20
    turns 3, 7, 12, 18

NEED_CODE: <short description of what to compute>

Response:
"""
```

Trajectory Based Code Generation Prompt Template

```
CODE_GENERATION_PROMPT_TEMPLATE = """You are helping to extract relevant
information from a trajectory to answer a question by writing Python code.

**Question:** {query}

**Task:** {task}

**Trajectory Format Reference (first 2 turns):**
{trajectory_sample}

**Trajectory Data (JSON format):**
Available as variable 'trajectory_json' with structure:
{
  "trajectory": [
    {
      "turn_idx": 0,       // Turn number (int)
      "action": "...",     // Action taken at this turn (string)
      "observation": "..." // Environment observation after action (string)
    },
    ...
  ],
  "task": "...",         // Task description (string)
  "episode_id": "..."    // Episode identifier (string)
}

**Your Task:**
Write Python code that processes the trajectory JSON and extracts the
relevant information to answer the question.
Example 1: Finding specific actions
```python
import json

trajectory_data = json.loads(trajectory_json)
trajectory = trajectory_data['trajectory']

relevant_turns = []
for turn in trajectory:
    if 'pick up' in turn.get('action', '').lower():
        relevant_turns.append({
            'turn': turn['turn_idx'],
            'action': turn['action'],
            'observation': turn.get('observation', '')[:200]
        })

result = {
    'relevant_turns': relevant_turns,
    'count': len(relevant_turns)
}
```

**Instructions:**
1. Write Python code that processes the trajectory JSON (available as
   variable 'trajectory_json')
2. Extract information relevant to answering the question
3. The code should be self-contained and executable
4. Store the final result in a variable named 'result'
**Output Format:**
**CODE**:
```python
# Your Python code here
```

Important: The code must be wrapped with **CODE**: marker followed by
```python code block.
"""
```

# L. Dataset Examples

We provide representative examples from subset below.

## L.1. Real-world subset example

---

**WebArena Case Study (ID: `task_webarena_17`)**

**Task:** Star the eight most-starred repositories on GitLab.
**Total Tokens:** 38,289    **Turns:** 31

| Turn | Action | Environment Observation (Structural State Snippet) |
|---|---|---|
| 00 | `click [426]` | Tab 0: `Projects · Dashboard · GitLab` (link `Explore`) |
| 01 | `click [2040]` | Tab 0: `Projects · Explore · GitLab` (link `Most stars`) |
| 04 | `click [6170]` | Tab 0: `Projects · Explore · GitLab` (link `267`) |
| 05 | `click [7798]` | Tab 0: `Issues · Umano: News Read To You / AndroidSlidingUpPanel` |
| ... | ... | ... |
| 15 | `type [13642]` | Tab 0: `most starred repositoriesstars · Search · GitLab` |
| ... | ... | ... |
| 23 | `type [19543]` | Tab 0: `The A11Y Project / allyproject.com` (textbox `Search`) |
| 26 | `click [21429]` | Tab 0: `Projects · Dashboard` (link `Byte Blaze / ally...`) |
| ... | ... | ... |
| 30 | `action: stop` | *[Early stop: Reach max steps 30]* |

**QA Examples:**

- **Type A (Temporal Inference):** *The search query string begins concatenating across Steps 14–16. At which step does the title first show 'most starred repositoriesstars', and what action caused that?*
  **Ans:** Step 15 shows the title, caused by the Step 14 `type` action appending text to the query without clearing it.

- **Type B (State Dependency):** *Why did keyword searching fail to replicate the 'Most stars' ordering, and what prerequisite state was required instead?*
  **Ans:** 'Most stars' is a view-specific ordering on Explore; Search matches text and returned zero results. The prerequisite was staying on Explore with 'Most stars' selected.

- **Type C (State Update):** *Compare the 'Users' result count between Step 14 and Step 21. What is the change?*
  **Ans:** It increases from 'Users 0' to 'Users 1', a +1 change.

- **Type D (State Abstraction):** *Identify the redundant loop near the end and explain why it is unproductive.*
  **Ans:** Steps 26 and 29 both click link [21429] from the Dashboard list without a state change in between. This loop fails to advance toward the goal.

---

## L.2. Synthetic subset example

---

### BabyAI Example (ID: `task_babyai_22`)

**Task:** put the purple key next to the green key
**Total Tokens:** 19,310    **Turns:** 392    **Result:** *fail (truncated)*

| Turn | Action | Environment Observation (Structural State Snippet) |
|---|---|---|
| 00 | done | Mission: put the purple key next to the green key. You are facing south. |
| 03 | toggle | In your view: . Walls border the area. You are facing south. |
| 10 | forward | In your view: a purple key, a red ball, a grey box. You are facing east. |
| 12 | toggle | In your view: a green key, a grey ball, a red ball, a grey box, a purple box, a purple key. You are facing north. |
| ... | ... | ... |
| 24 | pickup | In your view: a red ball, a grey box, a purple box, a purple key, a green box. You are facing north. |
| 43 | drop | In your view: . Walls border the area. You are facing south. |
| ... | ... | ... |
| 391 | toggle | Mission: put the purple key next to the green key. *[Reach max steps 392]* |

**QA Examples:**

- **Type A (Temporal Inference):** *Which object is no longer visible after the agent moves at step 10?*
  **Ans:** After the agent executes the 'forward' action at step 10, '. Walls border the area' is no longer visible. It was visible at step 9, but the agent's movement changed its field of view, causing '. Walls border the area' to move out of sight.

- **Type B (State Dependency):** *Background: At step 6, the agent's inventory is empty. Question: At step 7, can the agent execute the 'drop' action? Why or why not?*
  **Ans:** At step 7, the agent cannot execute the 'drop' action because the inventory is empty at step 6. The agent needs to have an item in inventory to drop it.

- **Type C (State Update):** *What are the inventory changes from step 1 to step 4? At which steps did the inventory change respectively?*
  **Ans:** The inventory changes from step 1 to step 4 are: step 4: agent picks up item (inventory changes).

- **Type D (State Abstraction):** *Summarize the agent's trajectory from step 0 to step 3 for the task 'put the purple key next to the green key'. What are the key actions and milestones?*
  **Ans:** From step 0 to step 3, the agent's trajectory includes: Key actions: step 0: done; step 1: done; step 2: done; step 3: toggle.

TextWorld Example (ID: `cooking_medium_103008`)

**Task:** Get ready to pick stuff up and put it in places... First thing I need you to do is to doublecheck that the walnut gate in the serious workshop is wide open. And then, go to the east... retrieve the old lampshade from the rusty rack... sit the old lampshade on the shoddy table inside the cramped vault.

**Total Tokens:** 8,384  **Turns:** 80  **Result:** *fail (reached_max_steps)*

| Turn | Action | Environment Observation (Structural State Snippet) |
|---|---|---|
| 00 | put old lampshade on... | You can't see any such thing. (Inventory: tiny loaf of bread, teal keycard, mouse, synthesizer) |
| 14 | look | **-= Serious Workshop =-** Desk contains outmoded mouse and tiny loaf of bread. Closed walnut gate to the east. |
| 29 | open walnut gate | You open walnut gate. (Available actions: go east, close walnut gate, examine walnut gate, etc.) |
| 59 | go east | **-= Cramped Vault =-** You see a shoddy table (empty) and a rusty rack with an **old lampshade** on it. |
| 60 | put old lampshade on... | You need to take the old lampshade first. |
| 64 | put old lampshade on... | You need to take the old lampshade first. |
| 74 | go north | **-= Silent Cubicle =-** The agent moves away from the goal room. |
| 78 | go south | **-= Serious Workshop =-** Agent returns to the starting room. |
| 79 | look | *[Early stop: Reach max steps 80]* |

**QA Examples (Direct Extraction):**

- **Type A (Temporal Inference):** *What actions were performed between step 37 and step 41?*
  **Ans:** Between step 37 and step 41, the agent performed the following actions: at step 37, eat tiny loaf of bread; at step 38, put old lampshade on shoddy table; at step 39, put old lampshade on shoddy table; at step 40, put old lampshade on shoddy table; at step 41, look.

- **Type B (State Dependency):** *At step 27 with state: Inventory: empty... Object states: shoddy table [cramped vault], old lampshade [rusty rack], teal keycard [I]... can the agent perform "drop teal keycard"?*
  **Ans:** No, the agent cannot perform this action because the inventory is empty, so there is no object to put.

- **Type C (State Update):** *How did the state of I change throughout the trajectory, including what objects were placed in or removed from it?*
  **Ans:** I evolution: step 1: tiny loaf of bread was removed; step 2: outmoded mouse was removed; step 15: tiny loaf of bread was added; step 23: tiny loaf of bread was removed; step 31: small synthesizer was removed; step 32: tiny loaf of bread was added; step 37: tiny loaf of bread was removed. At step 41, it contains: teal keycard.

- **Type D (State Abstraction):** *Until step 79, what actions has the agent performed and how frequently?*
  **Ans:** Actions performed: 'put old lampshade on shoddy table' (18 times), 'inventory' (13 times), 'look' (8 times), 'take old lampshade from rusty rack' (6 times), 'open walnut gate' (5 times), 'put tiny loaf of bread on solid desk' (4 times), 'take tiny loaf of bread from solid desk' (3 times), 'go east' (3 times), 'examine small synthesizer' (3 times), 'put outmoded mouse on solid desk' (2 times), 'examine teal keycard' (2 times), 'examine solid desk' (2 times), 'close walnut gate' (2 times), 'examine walnut gate' (1 times), 'drop small synthesizer' (1 times).

