# OpenReview forum: "AMA-Bench: Evaluating Long-Horizon Memory for Agentic Applications"
_ICML.cc/2026/Conference — ICML 2026 regular_

### Official Review · Reviewer_iTfx · 2026-03-09

**Soundness:** 3
**Presentation:** 2
**Significance:** 3
**Originality:** 4
**Overall Recommendation:** 4
**Confidence:** 3

**Summary:**

This paper introduces AMA-Bench, a benchmark for evaluating long-horizon memory in real agentic applications. The authors argue that existing memory benchmarks are largely dialogue-centric and therefore mismatch real-world agent trajectories, which (1) contain diverse machine-generated symbolic representations (e.g., JSON/HTML), (2) are causally grounded because each action induces environment state transitions that constrain future observations, and (3) consist of objective information, unlike dialogues that often include redundant chit chat. To bridge this gap, AMA-Bench is built from two complementary subsets: (i) a real-world subset, and (ii) a synthetic subset. Through a systematic evaluation of many memory systems on AMA-Bench, the results indicate that memory design is the primary bottleneck in long-horizon agent performance. Motivated by these findings, the authors propose AMA-Agent, which constructs a Causality Graph and Tool-Augmented Search. Overall, AMA-Bench offers a realistic and scalable testbed for diagnosing memory failures in real agent trajectories, and AMA-Agent provides a concrete direction for causality-aware memory architectures.

**Compliance With Llm Reviewing Policy:**

Affirmed.

**Final Justification:**

This concern has been sufficiently addressed through the authors’ clarifications and additional analysis, which resolve the initial ambiguity and make the argument more convincing.

**Key Questions For Authors:**

1. Can you report end-to-end task success for each baseline as well as AMA-Agent (at least on a subset where this is feasible)?
2. Alternatively, on the synthetic subset, could you increase the horizon length and report how end-to-end task success changes as QA performance changes (e.g., success vs. horizon, alongside QA accuracy vs. horizon)?
3. Alternatively, could you provide qualitative case studies showing that better long-horizon memory reduces unnecessary repetitive actions (or improves recovery from errors), to better illustrate how memory improvements translate to agentic behavior?
4. Could you add a quantitative analysis of the QA set quality, i.e., whether the QA pairs truly stress the long-horizon memory challenges you claim to target (e.g., evidence distance)?
5. I think the paper would benefit from a discussion of the cost and latency of AMA-Agent’s memory pipeline, particularly the overhead introduced by its memory construction and memory retrieval processes.

**Limitations:**

yes

**Strengths And Weaknesses:**

Strength
- Problem formulation. The paper clearly identifies that existing benchmarks for evaluating memory systems are largely dialogue-centric, and therefore fail to reflect key properties of real-world agent trajectories, such as representational diversity, causality, and dense objective information. This motivates a well-defined problem setting aimed at closing the gap between prior benchmarks and real-world agentic applications.
- Benchmark design. The benchmark is carefully constructed. The authors build a real-world subset and additionally introduce a synthetic subset, enabling controllable environments where difficulty and horizon length can be systematically scaled and tested.

Weakness
W1. Bridging Trajectory QA to End-to-End Agentic Success
- Since the evaluation is primarily trajectory log-based QA, there are inherent limitations in interpreting how these results relate to end-to-end agentic task success. To be clear, the authors do not claim that this benchmark directly guarantees agentic success. Still, the ultimate goal behind designing better memory systems is likely to improve success rates in end-to-end agentic tasks. Therefore, while this may fall slightly outside the paper’s intended scope, providing some evidence or discussion of how QA performance correlates with task success would make the benchmark’s necessity and impact more compelling.

W2. Reliability of the Subset
- In Figure 6, the long-context baseline outperforms most memory-based baselines, and the authors interpret this as evidence that existing memory systems suffer from losses introduced by compression and retrieval.
- However, it is also possible that the real-world subset is not consistently composed of QA pairs that are well-suited for evaluating long-horizon memory.
- For instance, a more targeted long-horizon memory stress test would include questions where the key evidence is located primarily in the early portion of the trajectory, or where answering requires aggregating evidence from multiple locations.
- While the paper provides many insightful analyses and ablations, it seems less clear whether the authors quantitatively demonstrate that the real-world subset is tightly aligned with the specific long-horizon memory challenges they aim to evaluate.

---

> ### Author Rebuttal · Authors · 2026-03-31
>
> We sincerely thank the reviewer recognizes the importance of moving beyond dialogue-centric memory benchmarks, the careful design of AMA-Bench, and the broader significance of studying long-horizon memory in realistic agent trajectories. We understand the main concerns to be whether the benchmark QA sets truly stress long-horizon memory, and how the observed QA gains relate to end-to-end agentic success. We address these points below.
>
>
>
> ### W1, Q1-Q3
> We agree that trajectory QA is not identical to end-to-end agentic success, and we do not intend to claim otherwise. Our goal in this paper is to isolate and evaluate **memory ability itself**, since memory is often entangled with many other factors in full agent execution.
>
> At the same time, we agree that downstream task success is the ultimate motivation for better memory systems. To address this, we are currently running **end-to-end execution baselines** on **BabyAI** and **TextWorld**, and will include these results in the revision if available. We will also clarify in the paper that end-to-end execution is an important complementary validation beyond retrieval and QA.
>
> ### W2, Q4
>
> For the **real-world subset**, QA pairs are authored by **graduate-level annotators with research experience in LLM agents**, following shared guidelines that standardize **evidence grounding** and **category coverage** across domains. Each annotated trajectory further undergoes a **cross-review sanity check by a second annotator**, which improves reliability and consistency.
>
> For the **synthetic subset**, we quantify whether questions truly require long-horizon reasoning rather than short local context:
>
> | Metric | Overall | BABYAI | TEXTWORLD |
> |---|---:|---:|---:|
> | Mean normalized evidence position | **0.424** | 0.476 | 0.420 |
> | Mean hop count | **4.22** | 11.39 | 3.64 |
> | Multi-hop ratio | **62.6%** | 100.0% | 59.7% |
> | Mean evidence span (turns, multi-hop only) |  | **73.7** | **9.3** |
> | Mean evidence span (tokens, multi-hop only) |  | **3,918** | **2,110** |
> | Mean normalized span (multi-hop only) |  | 0.137 | **0.302** |
>
> These results show that the subset is not dominated by trivial local questions. A large fraction of QA pairs require **multi-hop evidence aggregation**, and the supporting evidence often spans long portions of the trajectory.
>
> We also analyze **position coverage** across the full trajectory:
>
> | Split | First 20% | Middle 60% | Last 20% |
> |---|---:|---:|---:|
> | BABYAI | 9.3% | **89.4%** | 1.3% |
> | TEXTWORLD | 15.2% | **79.7%** | 5.0% |
>
> This shows that QA evidence is broadly distributed rather than concentrated only at the very end. Together, these results support that AMA-Bench does target the long-horizon memory challenges discussed in the paper.
>
>
> ### Q5
> We agree that a quantitative latency and cost analysis is important. We therefore add the following comparison on the real-world subset:
>
> | Method | Avg. Accuracy | Memory Construct Latency | Retrieve + Answer Latency | End-to-End Latency |
> |:--|--:|--:|--:|--:|
> | Mem0 | 0.2104 | 1.12 | 1.732 | 2.852 |
> | MemAgent | 0.2768 | 4.86 | 0.68 | 5.54 |
> | **AMA-Agent** | **0.5722** | 6.80 | 1.70 | 8.50 |
> | HippoRAG2 | 0.4480 | 10.90 | 1.20 | 12.10 |
> | Mem1 | 0.1229 | 21.57 | 0.46 | 22.03 |
> | SimpleMem | 0.1811 | 23.55 | 0.80 | 24.35 |
>
> These results show that AMA-Agent does introduce additional overhead relative to lightweight baselines, but the trade-off is favorable. In particular, AMA-Agent achieves the **best accuracy overall** while remaining substantially more efficient than several stronger memory systems such as **HippoRAG2, Mem1, and SimpleMem**. We will include this analysis in the revision and add token cost statistics for completeness.

---

> > ### Author Rebuttal · Reviewer_iTfx · 2026-04-03
> >
> > Thank you for the detailed response.
> >
> > Regarding W1, I understand that the main goal of this work is to isolate and evaluate memory ability itself, rather than to optimize directly for end-to-end downstream success. However, I still think it is important to understand whether evaluating memory only through QA pairs can meaningfully indicate whether that memory ultimately helps downstream tasks. This is why I raised the question. I appreciate that the authors agree this is an important direction, and I look forward to seeing the additional results.
> >
> > Regarding W2, I appreciate the deeper analysis provided for the synthetic subset. The evidence distribution and multi-hop statistics are helpful, and they make it more convincing that the synthetic subset is not dominated by trivial local questions. For the real-world subset, I understand that it was carefully constructed through human annotation and cross-review. However, I am not sure why the same kind of analysis was not also applied to the real-world subset. Is there a reason for this?
> >
> > Thank you again for the careful and thoughtful response. Since the authors mention that additional experiments are currently in progress, I do not think I can raise my score at this time. However, I would be open to revisiting the score if the updated results address these points.

---

> > > ### Author Response · Authors · 2026-04-07
> > >
> > > ### Q1: End to end results on feasible subsets
> > >
> > > We added end to end results on both a synthetic environment, **TextWorld**, and a real world setting, **Spider2** (text to SQL). To ensure a controlled comparison, we use the same vanilla **Qwen3 32B** execution agent for all methods and vary only the memory mechanism.
> > >
> > > | Method      | TextWorld E2E | TextWorld QA | Spider2 E2E | Spider2 QA |
> > > | ----------- | ------------: | -----------: | ----------: | ---------: |
> > > | AMA Agent   |          51.5 |       40.434 |        26.2 |      57.38 |
> > > | LongContext |          47.5 |         33.0 |        23.5 |      50.86 |
> > > | HippoRAG    |          27.5 |         11.6 |        23.8 |      50.49 |
> > > | Embedding   |          37.5 |         13.9 |        21.4 |      45.42 |
> > > | MemoryBank  |          31.5 |         11.9 |        17.2 |      25.98 |
> > > | Mem0        |          30.0 |         11.8 |        15.6 |      12.93 |
> > >
> > > Across methods, QA accuracy and end to end success are strongly correlated in both settings. We observe three main patterns:
> > > (1) **Agentic Compressed memory is lossy**, often dropping precise environmental states and long range dependencies.
> > > (2) **Similarity based retrieval is fragile** in strongly causal environments such as TextWorld.
> > > (3) **AMA Agent has a structural advantage**: it preserves information fidelity while supporting graph based and tool based search, which better retains causal dependencies.
> > >
> > > #### Efficiency comparison on representative methods
> > >
> > > | Method      | Avg latency / QA | Avg latency / task | Max latency / QA | Max latency / task |
> > > | ----------- | ---------------: | -----------------: | ---------------: | -----------------: |
> > > | LongContext |              1.4 |               2176 |              2.2 |               2932 |
> > > | HippoRAG    |             12.1 |               3660 |             47.2 |               7290 |
> > >
> > > QA evaluation is scalable and largely parallelizable, making it much more efficient than full end to end rollouts. By contrast, an end to end agent must interact with the environment in a strictly sequential loop, often for **80 to 100 steps** per task, with repeated agent environment interactions. This makes **per task latency much larger**. Moreover, end to end evaluation makes it harder to isolate whether a failure comes from memory quality or downstream execution errors. This is why **QA based evaluation** is useful: it isolates memory quality in a more scalable and controlled way.
> > >
> > > ---
> > >
> > > ### Q2: TextWorld end to end success by difficulty
> > >
> > > | Method      | Easy | Medium | Hard | Very hard | E2E |
> > > | ----------- | ---: | -----: | ---: | --------: | --: |
> > > | AMA Agent   |   84 |     61 |   35 |        22 | 50.5 |
> > > | LongContext |   85 |     55 |   32 |        18 | 47.5 |
> > > | HippoRAG    |   63 |     30 |   12 |         5 | 27.5 |
> > > | Embedding   |   68 |     48 |   26 |         8 | 37.5 |
> > > | MemoryBank  |   53 |     36 |   23 |        14 | 31.5 |
> > > | Mem0        |   50 |     35 |   20 |        12 | 30.0 |
> > >
> > > These results suggest that agent performance is highly memory sensitive, and that AMA Agent degrades more gracefully as difficulty and horizon increase.
> > >
> > > ---
> > >
> > > ### Q3: Qualitative case studies on agentic behavior
> > >
> > > We also include concise TextWorld style case studies here.
> > >
> > > | Behavior category | RAG based | Agentic memory | Causality graph |
> > > |---|---|---|---|
> > > | **Reduced repetitive actions** | Retrieves semantically related clues such as *antidote* or *alchemy room* but misses the prerequisite chain **retrieve key -> unlock basement -> access alchemy room**, leading to repeated invalid actions such as retrying `open door`. | Compresses the trajectory into a coarse goal such as *craft antidote in basement* while omitting executable preconditions, which also leads to repetitive futile behavior. | Explicitly preserves action dependencies, so the agent first retrieves the key, unlocks the basement, and then crafts the antidote. |
> > > | **Improved recovery from errors** | After an early mistake, retrieval may still surface stale but relevant fragments, causing the agent to keep following an invalid plan. | A compressed summary may preserve the original plan but fail to reflect an irreversible state change. | Explicitly updates the consequence chain, allowing the agent to abandon the invalid branch and recover appropriately. |
> > >
> > > Overall, RAG based memory tends to fail by retrieving semantically relevant but causally incomplete evidence, summary based memory tends to fail by compressing away critical preconditions and state changes.
> > >
> > > ---
> > >
> > > ### Q4: Additional analysis of evidence position
> > >
> > > We also analyzed evidence position for the realworld subset:
> > >
> > > | Split | 0-20% | 20-60% |60-100% |
> > > |---|---:|---:|---:|
> > > | BABYAI | 9.3% | 89.4% | 1.3% |
> > > | TEXTWORLD | 15.2% | 79.7% | 5.0% |
> > > This shows that supporting evidence is broadly distributed across the trajectory rather than concentrated only at the very end. Additional real world analysis is provided in our second round response to **Q4 for JgyK**.

---

### Official Review · Reviewer_JgyK · 2026-03-12

**Soundness:** 3
**Presentation:** 3
**Significance:** 3
**Originality:** 3
**Overall Recommendation:** 5
**Confidence:** 3

**Summary:**

This paper introduces AMA-Bench, a benchmark for evaluating long-horizon memory in realistic agentic applications rather than dialogue-only settings. The benchmark includes both real-world agent trajectories with expert-curated QA and synthetic trajectories with controllable horizon length and rule-based QA. Based on the finding that existing memory systems often lose causal and objective information in long trajectories, the authors also propose AMA-Agent, a memory system built on a causality graph and tool-augmented retrieval. Experiments show that AMA-Agent outperforms prior memory baselines by a notable margin on AMA-Bench.

**Compliance With Llm Reviewing Policy:**

Affirmed.

**Final Justification:**

Thanks for the response that resolves my concern. If all the additional experiments have been added to the paper, the paper is solid. I have increased my score.

**Key Questions For Authors:**

1. How much of AMA-Agent’s gain comes from the causality graph itself versus tool-augmented retrieval? A stronger controlled comparison would clarify whether the improvement mainly comes from memory structure, tool use, or both.
2. Can the authors compare AMA-Agent against stronger or more recent memory and retrieval baselines under similarly optimized settings? This would help verify that the gains are not mainly due to weaker baseline choices.
3. How well does AMA-Agent generalize beyond AMA-Bench? Results on another benchmark or downstream agent task would strengthen the claim that the method captures a broader principle rather than a benchmark-specific design.
4. More evidence on annotation quality and category difficulty would strengthen confidence in the benchmark.

I would increase my score if my concerns were well addressed.

**Limitations:**

yes

**Strengths And Weaknesses:**

The paper's main strength is the clear shift from dialogue-centric memory evaluation to more realistic agent-environment interaction settings. AMA-Bench appears meaningful and well-motivated, and the combination of real-world and synthetic data makes the benchmark both practical and scalable. The proposed AMA-Agent is also intuitively aligned with the benchmark’s challenges, and the empirical results are promising. However, the method is somewhat tightly matched to the benchmark’s framing, making it harder to judge how broadly the gains generalize beyond AMA-Bench. In addition, while the reported improvement over memory baselines is strong, the absolute performance still leaves substantial room for improvement, and stronger comparisons against more recent or better-optimized baselines would make the evaluation more convincing.

---

> ### Author Rebuttal · Authors · 2026-03-31
>
> We sincerely thank the reviewer for the positive assessment of our work. We are encouraged that the reviewer recognizes the practical value, and significance of shifting memory evaluation from dialogue only settings to realistic agent-environment interactions.  The main concerns, as we understand them, are whether the observed gains generalize beyond AMA-Bench, whether the baseline comparisons are sufficiently strong, and whether the benchmark quality is supported by enough evidence. We address these points below.
>
> ## Response to Key Questions
>
> ### Q1
> Tab. 6 in the paper suggests that AMA-Agent’s gain is not driven by only one component. Removing the **causality graph** reduces the average score from **0.57** to **0.43**, while removing **tool-augmented retrieval** reduces it to **0.44**. This shows that the two components contribute comparably overall, with the causality graph being slightly stronger on average. In an additional analysis over 200 randomly sampled QA pairs, AMA-Agent invoked graph node search in 10.0% of cases and keyword tools in 23.5% of cases.
>
> ### Q2
> We appreciate the reviewer’s suggestion regarding stronger baselines. In our experiments, we already compare against recent and competitive memory systems, like the latest work **SimpleMem**, a January 2026 method. To ensure fairness, we reproduced baselines using the optimized settings recommended by their original papers, as described in Sec. 6.1 and Appendix B.
>
> Therefore, we believe the gains of AMA-Agent are not due to weak baseline configurations, but reflect the advantage of the proposed architecture under competitive settings.
>
> ### Q3
> To evaluate generalization beyond AMA-Bench, we additionally test AMA-Agent on the established **LoCoMo** benchmark using **Qwen3-32B** as the base model:
>
> | Method | LoCoMo (Qwen3-32B) |
> |:--|--:|
> | A-mem | 60.84 |
> | Mem0 | 63.51 |
> | GraphRAG | 65.39 |
> | HippoRAG | 72.10 |
> | **AMA-Agent** | **76.20** |
>
> AMA-Agent outperforms all compared baselines, surpassing the strongest baseline, **HippoRAG**, by **4.1 points**. We believe this result supports that the method captures a more general principle for long-horizon memory, rather than being narrowly tailored to AMA-Bench.
>
> ### Q4
> We agree that benchmark confidence should be supported by stronger evidence on annotation quality and category difficulty.
>
> For the **real-world subset**, QA pairs are authored by **graduate-level annotators with research experience in LLM agents**, following shared guidelines that standardize **evidence grounding** and **category coverage** across six domains. To improve annotation reliability, each annotated trajectory further undergoes a **cross-review sanity check by a second annotator**. This design makes the QA labels grounded, auditable, and consistent across domains.
>
> For the **synthetic subset**, we control data quality from three aspects:
>
> #### a. Answer correctness
> Synthetic QA pairs are grounded in **deterministic environment backends** rather than LLM generated answers. As described in Sec. 3.3.2, answers are programmatically extracted from environment states and execution logs, which eliminates hallucination risk. We also conduct expert manual spot checks to verify correctness.
>
> #### b. Distributional quality
> We verify that evidence is distributed across the trajectory, rather than concentrated only at the beginning or the end. Using **20% bins**, the evidence distribution is:
>
> | Range | Turn % | Token % |
> |:--|--:|--:|
> | 0% to 20% | 15.1% | 15.7% |
> | 20% to 40% | 26.8% | 27.3% |
> | 40% to 60% | 42.8% | 41.6% |
> | 60% to 80% | 10.3% | 10.7% |
> | 80% to 100% | 5.0% | 4.8% |
>
> This shows that the supporting evidence is broadly distributed, with the largest mass in the middle of the trajectory rather than at trivial positions.
>
> #### c. Real-world consistency
> The synthetic subset is also predictive of real-world behavior. Empirically, we observe consistent trends across models and categories, which suggests that it captures meaningful long-horizon memory challenges rather than synthetic artifacts.
>
> In addition, category difficulty is stable across models. Multiple methods show a similar ordering, with **Recall** and **Causal Inference** generally easier than **State Updating** and **State Abstraction**. We view this consistency as evidence that the category definitions reflect meaningful differences in reasoning difficulty rather than annotation noise.

---

> > ### Author Rebuttal · Reviewer_JgyK · 2026-04-04
> >
> > Thanks for the answer.
> >
> > The ablation analysis (Q1) and the LoCoMo generalization result (Q3) are helpful and address those concerns. However, for Q2, the response mainly restates that existing baselines were faithfully reproduced rather than adding stronger comparisons. For Q4, the annotation quality evidence is primarily provided for the synthetic subset; quantitative analysis for the real-world subset (e.g., evidence distance, multi-hop statistics) is still missing, and are there any stats in your reviewing process that can be revealed as proof?

---

> > > ### Author Response · Authors · 2026-04-07
> > >
> > > ### For Q2,
> > > The main reason is that this analysis requires additional human annotation. To address this concern, we now provide an additional study on the real-world subset by randomly sampling 120 QA pairs from each of 6 tasks, yielding 720 QA pairs** in total out of 2,496 candidate QA pairs (28.8% of the full real-world subset).
> > >
> > > #### Long-Horizon Reasoning Metrics on the Real-World Subset
> > >
> > > | Metric | Overall |
> > > |---|---:|
> > > | Mean normalized evidence position | 0.475 |
> > > | Mean hop count | 3.38 |
> > > | Multi-hop ratio | 72.2% |
> > > | Mean evidence span (turns, multi-hop only) | 16.1 |
> > > | Mean normalized span (multi-hop only) | 0.260 |
> > >
> > > ### Position Coverage Across the Full Trajectory
> > >
> > > | Split | [0%, 20%) | [20%, 80%) | [80%, 100%] |
> > > |---|---:|---:|---:|
> > > | Overall | 21.4% | 62.9% | 15.7% |
> > >
> > > These results show that evidence in the real-world subset is also broadly distributed across the trajectory and frequently requires multi-hop reasoning.
> > >
> > >
> > > #### For Q4,
> > >
> > > To address this concern, we added experiments with two recent methods under the same **Qwen3-32B** agent setup: **HiMem**[1] (**Jan. 2026**) and **EMem**[2] (**Nov. 2025**). Both were released very close to the **ICML submission deadline**, and thus represent near deadline, state of the art baselines for this comparison.
> > >
> > > | Method | Recall | Causal Inference | State Updating | State Abstraction | Average |
> > > |---|---:|---:|---:|---:|---:|
> > > | HiMem | 0.2945 (0.2241) | 0.4018 (0.2065) | 0.3126 (0.1932) | 0.2238 (0.2087) | 0.2995 (0.2198) |
> > > | EMem | 0.4631 (0.2412) | 0.4925 (0.1878) | 0.4512 (0.1963) | 0.3421 (0.1824) | 0.4610 (0.2095) |
> > > | AMA-Agent (AMA) | **0.6238 (0.3280)** | **0.6145 (0.3103)** | **0.5305 (0.2625)** | **0.4719 (0.2825)** | **0.5722 (0.2992)** |
> > >
> > > In addition, for **HippoRAG** and **MemoRAG**, we further tuned their key hyperparameters to test whether the gap could be explained by suboptimal settings. The effect of such tuning is limited.
> > >
> > > **HippoRAG, probability factor**
> > >
> > > | Weight | 0.01 | 0.05 | 0.1 | 0.3 |
> > > |---|---:|---:|---:|---:|
> > > | Score | 0.429 | 0.457 | 0.448 | 0.440 |
> > >
> > > **MemoRAG, compression ratio**
> > >
> > > | Compression Ratio | 2 | 4 | 8 | 16 |
> > > |---|---:|---:|---:|---:|
> > > | Score | 0.4729 | 0.4604 | 0.4302 | 0.4100 |
> > >
> > > Overall, these results suggest that the improvement of **AMA-Agent** is unlikely to be explained by weak baselines or insufficient tuning. Rather, the main difference appears to come from the underlying memory method.

---

### Official Review · Reviewer_jPyg · 2026-03-12

**Soundness:** 2
**Presentation:** 3
**Significance:** 3
**Originality:** 3
**Overall Recommendation:** 4
**Confidence:** 4

**Summary:**

This paper highlights a critical bias in the current evaluation of memory for Large Language Models (LLMs) in agentic applications: existing memory benchmarks predominantly focus on dialogue-centric interactions, whereas real-world agent scenarios consist of long-horizon, machine-generated representations with strong causal dependencies. To bridge this gap, the authors propose AMA-Bench for evaluating memory in agent applications. Through extensive systematic evaluations, the authors find that existing memory systems often underperform when processing high-density objective information. Motivated by this, the paper introduces AMA-Agent, which incorporates a Causality Graph and Tool-Augmented Retrieval, achieving performance that significantly surpasses existing baselines on AMA-Bench.

**Compliance With Llm Reviewing Policy:**

Affirmed.

**Final Justification:**

I appreciate the authors' effort in conducting the new experiments. My concerns are fully resolved, and I am increasing my score to Weak Accept.

**Key Questions For Authors:**

**Q1:** Regarding the enhancement of long-horizon memory for LLMs in real agentic applications: Does this memory improvement directly translate to better performance on the corresponding downstream agent execution benchmarks? To what extent does this enhanced memory capability actually improve the agent's end-to-end task success rate?

**Q2:** Could you provide more details on how the Causality Graph is constructed in practice? In real-world Web environments (e.g., HTML containing massive amounts of irrelevant DOM nodes) or Software Engineering environments, state extraction is notoriously difficult. How does the system ensure the correctness and completeness of this extraction? Additionally, what are the specific time latency and token consumption (cost) required to construct such a dynamic graph?

**Limitations:**

yes

**Strengths And Weaknesses:**

### **Strengths**

- **S1:** The paper accurately characterizes the fundamental difference between "dialogue memory" and "agent trajectory memory." The insight that existing methods fail to handle agent environments (which contain dense machine-generated symbols and strict causal logic) due to their reliance on lossy compression and pure similarity-based retrieval is somewhat inspiring.
- **S2:** The experimental section effectively reveals that existing compression mechanisms and similarity-based retrieval have flaws when processing dense agent trajectories.

### **Weaknesses**

- **W1:** The paper does not adequately clarify how to robustly and accurately extract causal edges and state nodes from noisy and unstructured information, such as complex HTML DOM trees or large code repositories. Furthermore, it lacks a discussion on the potential cascading effects caused by erroneous extraction.
- **W2:** The paper lacks a quantitative trade-off analysis regarding the computational overhead, inference latency, and token cost versus the performance gains associated with the proposed Causality Graph and Tool-Augmented Retrieval.
- **W3:** The paper does not provide a sufficient discussion or evaluation of AMA-Agent's performance on actual, end-to-end real-world agent execution tasks (beyond just memory retrieval and QA).

---

> ### Author Rebuttal · Authors · 2026-03-31
>
> We sincerely thank the reviewer for the positive assessment of our work. We acknowleged that the reviewer recognizes the novelty, practical value, and significance of studying long-horizon memory for agentic systems. We understand that the main concerns are regarding graph construction details, latency and cost trade-offs, and end-to-end policy evaluation. We will clarify these points in the revision.
>
> ## Response to Weaknesses
>
> ### W1
> We thank the reviewer for raising this important point. In practice, our **Causality Graph** is not constructed through a strict rule based parser over raw HTML DOM trees or software states. Instead, following Sec. 5.1 and App. D.1, we prompt an LLM to analyze each local trajectory segment \((o_{t-1}, a_t, o_t)\) and extract **task relevant objects, state changes, and their latent causal dependencies** in a structured form.
>
> Therefore, the current implementation should be understood as an **LLM based semantic abstraction over trajectories**, rather than exact symbolic reconstruction of all low level environment states. For example, in a web task asking for the bird species featured in a BBC Earth video, the extracted graph does not preserve noisy page artifacts. Instead, it tracks higher level entities such as the **BBC Earth video**, **penguins**, the **search page**, and the **generated report**. We will clarify this design choice more explicitly in the revision and discuss the possible impact of extraction errors.
>
> ### W2
> We agree that a quantitative latency and cost analysis is important. We therefore add the following comparison on the real-world subset:
>
> | Method | Avg. Accuracy | Memory Construct Latency | Retrieve + Answer Latency | End-to-End Latency |
> |:--|--:|--:|--:|--:|
> | Mem0 | 0.2104 | 1.12 | 1.732 | 2.852 |
> | MemAgent | 0.2768 | 4.86 | 0.68 | 5.54 |
> | **AMA-Agent** | **0.5722** | 6.80 | 1.70 | 8.50 |
> | HippoRAG2 | 0.4480 | 10.90 | 1.20 | 12.10 |
> | Mem1 | 0.1229 | 21.57 | 0.46 | 22.03 |
> | SimpleMem | 0.1811 | 23.55 | 0.80 | 24.35 |
>
> These results show that AMA-Agent does introduce additional overhead compared with lightweight baselines, but the overhead is well justified by the performance gain. In particular, AMA-Agent achieves the **best accuracy overall** while maintaining a **more favorable latency-accuracy trade-off than stronger competing memory systems** such as HippoRAG2, Mem1, and SimpleMem. We will include this analysis in the revision and also add token cost statistics for completeness.
>
> ### W3
> We agree with the reviewer that the current paper does not yet provide sufficient evaluation of AMA-Agent on **end-to-end real-world agent execution tasks** beyond retrieval and QA.  We are currently running **end-to-end agent execution baselines** on BabyAI and textworld, and plan to add these results before the rebuttal period ends. We will also clarify in the revision that the downstream execution is an important complementary validation.
>
> ## Response to Key Questions
>
> ### Q1
> See W3.
>
> ### Q2
> See **W1** for the graph construction details. For latency and cost, see **W2**.

---

> > ### Author Rebuttal · Reviewer_jPyg · 2026-04-04
> >
> > Thank you for the detailed rebuttal. I will consider adjusting my score once the supplementary results for the end-to-end real-world agent execution tasks are provided.

---

> > > ### Author Response · Authors · 2026-04-07
> > >
> > > Thank you for the thoughtful follow up. In response, we have now added supplementary end to end results on both a synthetic environment, **TextWorld**, and a real world setting, **Spider2** (text to SQL), using the same vanilla **Qwen3 32B** execution agent across all methods and varying only the memory mechanism.
> > >
> > > | Method      | TextWorld E2E | TextWorld QA | Spider2 E2E | Spider2 QA |
> > > | ----------- | ------------: | -----------: | ----------: | ---------: |
> > > | AMA Agent   |          51.5 |       40.434 |        26.2 |      57.38 |
> > > | LongContext |          47.5 |         33.0 |        23.5 |      50.86 |
> > > | HippoRAG    |          27.5 |         11.6 |        23.8 |      50.49 |
> > > | Embedding   |          37.5 |         13.9 |        21.4 |      45.42 |
> > > | MemoryBank  |          31.5 |         11.9 |        17.2 |      25.98 |
> > > | Mem0        |          30.0 |         11.8 |        15.6 |      12.93 |
> > >
> > > Across methods, QA accuracy and end to end success are strongly correlated in both settings. We also observe that compressed memory is often lossy for precise state tracking, similarity based retrieval is fragile in strongly causal environments such as TextWorld, and AMA Agent benefits from preserving information fidelity while enabling graph based and tool based memory search.
> > >
> > > For efficiency, we additionally report representative latency numbers on TextWorld:
> > >
> > > | Method      | Avg latency / QA | Avg latency / task | Max latency / QA | Max latency / task |
> > > | ----------- | ---------------: | -----------------: | ---------------: | -----------------: |
> > > | LongContext |              1.4 |               2176 |              2.2 |               2932 |
> > > | HippoRAG    |             12.1 |               3660 |             47.2 |               7290 |
> > >
> > > These numbers further highlight why **QA based evaluation** is valuable: it is scalable and largely parallelizable, whereas end to end agent rollouts are strictly sequential, often requiring **80 to 100 interaction steps** per task, which makes per task latency much larger and failure attribution less clean.
> > >
> > > For more detailed discussion, including qualitative case studies and additional analysis, please see our responses to **iTfx Q1 to Q3**.

---

### Official Review · Reviewer_zU5b · 2026-03-13

**Soundness:** 3
**Presentation:** 2
**Significance:** 3
**Originality:** 3
**Overall Recommendation:** 4
**Confidence:** 3

**Summary:**

This paper introduces AMA-Bench, a meticulously curated benchmark for long-horizon agentic tasks. Distinct from prior works, this benchmark features diverse agent trajectories rather than being language-centric. It includes both real-world and synthetic subsets, covering domains such as web, open-world QA, Text2SQL, software engineering, gaming, and embodied AI. Based on evaluation results from this benchmark, the authors further propose AMA-Agent, which enhances long-term memory and retrieval capabilities through causality graphs and tool-augmented retrieval mechanisms. Experimental results on AMA-Bench demonstrate the effectiveness of the proposed method.

**Compliance With Llm Reviewing Policy:**

Affirmed.

**Final Justification:**

The authors' response has addressed my concerns. Overall, I have a positive view of this work and believe it will offer insights into long-horizon agentic tasks. I encourage the authors to consider exploring online evaluation, since it could provide a more direct assessment of end-to-end trajectory completion relative to offline protocols. I maintain my original positive score.

**Key Questions For Authors:**

The reviewer holds a positive view of this paper, with the main concerns centered around unclear benchmark and evaluation details.
1. Are all evaluations conducted in a QA format? Are the evaluation questions all open-ended, or are some multiple-choice with provided answer options？
2. Is the evaluation along each trajectory performed in an offline or online manner?  Do historical QA records influence the assessment of current steps?
3. How is the accuracy for the four dimensions in Table 4 (recall, causal inference, state updating, state abstraction) calculated?
4. Regarding the synthetic trajectories, how is data quality controlled and validated during generation?

**Limitations:**

Please see the Weaknesses and Key Questions sections for more details.

**Strengths And Weaknesses:**

Strength:
1. Long-horizon agentic tasks represent a rapidly evolving area of research, and the proposed benchmark offers a valuable reference for evaluating how well current agents handle such tasks, making it a meaningful contribution to the community.
2. Evaluations conducted on this benchmark provide critical insights, particularly highlighting the limitations of existing approaches in memory construction and retrieval.
3. Building on these insights, the paper introduces a well-motivated method to address the identified shortcomings, achieving strong performance on AMA-Bench.

Weakness:
1. Details regarding the benchmark construction and evaluation process are unclear. For instance, in Table 4, how are the four dimensions (recall, causal inference, state updating, and state abstraction) evaluated? Is each dimension assessed through corresponding QA tasks?
2. It is unclear how evaluations are conducted for each QA along the trajectory. Are QA pairs treated as independent instances and evaluated in an offline manner, or is evaluation performed in a streaming fashion where earlier QA content influences subsequent responses? A diagram illustrating the model's data flow would help readers better understand the evaluation protocol.
3. The agent is implemented using Qwen3-32B, and evaluation also relies on Qwen3-32B, which raises concerns about potential evaluation bias.
4. The analysis of AMA-Agent's two core components (i.e., Causality Graph and Tool-Augmented Retrieval) is insufficient. For example, details such as the types and frequency of tool usage are not discussed.
5. Experiments are primarily conducted on the proposed AMA-Bench. It would strengthen the paper to demonstrate the consistency and effectiveness of the method on other established memory evaluation benchmarks.

---

> ### Author Rebuttal · Authors · 2026-03-31
>
> We sincerely thank the reviewer for the positive assessment of our work. We are encouraged that the reviewer recognizes the novelty, practical value, and community relevance of studying long-horizon agent memory, as well as the strong performance of AMA-Agent on AMA-Bench. The main issues appear to be clarity of presentation and the need for additional empirical analysis. We will address these points in the revision.
>
> ## Response to Weaknesses
>
> ### W1
> Yes, each of the four dimensions is directly evaluated through corresponding QA tasks. Specifically, we designed targeted QA pairs that explicitly probe an agent's ability in recall, causal inference, state updating, and state abstraction based on its trajectory in the environment. The results presented in Tab. 4 represent the models' accuracy across these specific QA categories. We have provided concrete examples in Appendix E.1
>
> ### W2
> Our evaluation is conducted in an **offline** manner. Each QA pair is treated as an **independent instance** with the full trajectory as fixed context. Earlier questions do not affect later ones. This follows prior memory benchmarks such as **LoCoMo, SimpleMem, AMem, and MemoRAG**. We will clarify this protocol and add an illustration of the evaluation pipeline in the revision.
>
> ### W3
> To address possible evaluation bias, we added experiments using different model. We additionally use **Qwen3-8B** as base model on realworld. We also use **GPT-5 mini** from a different model family on synthetic set.
>
> | Method | Qwen3-32B| Qwen3-8B | GPT-5 mini (synthetic set) |
> |:--|--:|--:|--:|
> | Mem0 | 0.2095 | 0.2770 | 0.1141 |
> | HippoRAG2 | 0.4378 | 0.3836 | 0.0732 |
> | Mem1 | 0.1218 | 0.1429 | 0.0928 |
> | MemoryBank | 0.3417 | 0.2648 | 0.0725 |
> | A-mem | 0.3208 | 0.3498 | 0.1158 |
> | **AMA-Agent** | **0.5629** | **0.4814** | **0.2075** |
>
> AMA-Agent remains the strongest method across all settings.
>
>
> ### W4
> We agree that the analysis of the two core components can be made clearer.
>
> As described in Sec. 5.2 and Appendix D.3, AMA-Agent first performs **top K embedding retrieval**, then **self-evaluates** whether the evidence is sufficient. Only when it is insufficient does it invoke additional tools. The tool space includes:
>
> 1. **Graph node search**, which traverses the causality graph to gather multi-hop context and dependencies.
> 2. **Keyword tools**, where the agent writes and executes lightweight scripts for keyword matching or simple aggregation over the trajectory.
>
> In an additional analysis over **200 randomly sampled QA pairs**, AMA-Agent invoked **graph node search** in **10.0%** of cases and **keyword tools** in **23.5%** of cases.
>
> ### W5
> We additionally evaluate AMA-Agent on the established **LoCoMo** benchmark using Qwen3-32B as the base model.
>
> | Method | LoCoMo (Qwen3-32B) |
> |:--|--:|
> | A-mem | 60.84 |
> | Mem0 | 63.51 |
> | GraphRAG | 65.39 |
> | HippoRAG | 72.10 |
> | **AMA-Agent** | **76.20** |
>
> AMA-Agent outperforms all baselines and surpasses the strongest baseline,showing that the advantage of our method transfers beyond AMA-Bench.
>
> ## Response to Key Questions
>
> ### Q1
> Yes. In addition, all evaluation questions are **open-ended**, and we use **LLM-as-a-judge** for correctness. We report human agreement analysis in Appendix C.2. Also, we evaluate and found Qwen3-32B shows high agreement with other judge models:
>
> - vs. GPT-5.4: **92.8%**
> - vs. GPT-5.2: **91.8%**
> - vs. Claude-4.6: **88.8%**
> - vs. DeepSeek-v3.2: **84.7%**
>
> ### Q2
> See **W2**.
>
> ### Q3
> See **W1**.
>
> ### Q4
> We control synthetic data quality from three aspects:
>
> #### a. Answer correctness
> The synthetic QA pairs are grounded in **deterministic environment backends** rather than LLM generated answers. As described in Sec. 3.3.2, answers are programmatically generated based on the environment state and execution logs, which eliminates hallucination risk. We also perform expert manual spot checks to verify correctness.
>
> #### b. Distributional quality
> We verify the evidence distribution on the synthetic subset and found that evidence is distributed across the trajectory rather than concentrated only at the beginning or end. The evidence distribution is:
>
> | Range | Turn % | Token % |
> |:--|--:|--:|
> | 0% to 20% | 15.1% | 15.7% |
> | 20% to 40% | 26.8% | 27.3% |
> | 40% to 60% | 42.8% | 41.6% |
> | 60% to 80% | 10.3% | 10.7% |
> | 80% to 100% | 5.0% | 4.8% |
>
> This shows the evidence is broadly spread, with the largest mass in the middle rather than concentrated at trivial positions.
>
> #### c. Real-world consistency
> The synthetic subset is also predictive of real-world behavior. Empirically, we observe consistent trends across models and categories, suggesting that it captures meaningful long-horizon memory challenges rather than synthetic artifacts.

---

> > ### Author Rebuttal · Reviewer_zU5b · 2026-04-04
> >
> > Thank you for the detailed response and analysis, which address my concerns. I will maintain my positive score.

---

> > > ### Author Response · Authors · 2026-04-07
> > >
> > > Thank you very much for your constructive suggestions, and for maintaining your original positive score. We are grateful that our rebuttal addressed your main concerns, and we especially appreciate your suggestion on exploring online evaluation. We have also added supplementary end to end results on both a synthetic environment, **TextWorld**, and a real world setting, **Spider2** (text to SQL), using the same vanilla **Qwen3 32B** execution agent across all methods and varying only the memory mechanism:
> > >
> > > | Method      | TextWorld E2E | TextWorld QA | Spider2 E2E | Spider2 QA |
> > > | ----------- | ------------: | -----------: | ----------: | ---------: |
> > > | AMA Agent   |          51.5 |       40.434 |        26.2 |      57.38 |
> > > | LongContext |          47.5 |         33.0 |        23.5 |      50.86 |
> > > | HippoRAG    |          27.5 |         11.6 |        23.8 |      50.49 |
> > > | Embedding   |          37.5 |         13.9 |        21.4 |      45.42 |
> > > | MemoryBank  |          31.5 |         11.9 |        17.2 |      25.98 |
> > > | Mem0        |          30.0 |         11.8 |        15.6 |      12.93 |
> > >
> > > Across methods, QA accuracy and end to end success are strongly correlated in both settings. We also observe that compressed memory is often lossy for precise state tracking, similarity based retrieval is fragile in strongly causal environments such as TextWorld, and AMA Agent benefits from preserving information fidelity while enabling graph based and tool based memory search.
> > >
> > > For efficiency, we additionally report representative latency numbers on TextWorld:
> > >
> > > | Method      | Avg latency / QA | Avg latency / task | Max latency / QA | Max latency / task |
> > > | ----------- | ---------------: | -----------------: | ---------------: | -----------------: |
> > > | LongContext |              1.4 |               2176 |              2.2 |               2932 |
> > > | HippoRAG    |             12.1 |               3660 |             47.2 |               7290 |
> > >
> > > These results further motivate our use of **QA based evaluation**: QA can be evaluated scalably and largely in parallel, whereas end to end rollouts are strictly sequential, often requiring **80 to 100 interaction steps** per task, which makes per task latency much larger and failure attribution less clean.
> > >
> > > For more detailed discussion, including qualitative case studies and additional analysis, please see our responses to **iTfx Q1 to Q3**.

---

### Decision · Program_Chairs · 2026-04-30

**Decision:**

Accept (regular)

**Comment:**

I recommend Accept. The reviewer consensus is positive and there is little substantive disagreement. Reviewers generally agree that the paper identifies an important gap in existing memory benchmarks for LLM agents, namely that prior work is too dialogue-centric and does not adequately capture the long-horizon, causally structured, and information-dense nature of real agent trajectories. They also find AMA-Bench timely and useful, especially its combination of real-world and synthetic settings, and view AMA-Agent as a well-motivated method that achieves strong empirical results on the proposed benchmark.

The main concerns were not about the overall value of the paper, but about clarity and completeness: several reviewers asked for stronger explanation of the evaluation protocol, more analysis of the causality graph and tool-augmented retrieval components, stronger evidence of generalization beyond AMA-Bench, and more discussion of efficiency and end-to-end impact on downstream agent performance. In the rebuttal and follow-up discussion, the authors addressed these points with additional experiments and analyses, including results on LoCoMo, added comparisons to stronger baselines, efficiency analysis, and end-to-end results showing that improved memory quality correlates with downstream task success. These responses resolved the main reviewer concerns, and the final assessments remained uniformly positive.

Overall, I find this to be a solid and well-motivated paper with a useful benchmark contribution and a convincing accompanying method. While there is still room to further strengthen the end-to-end evaluation story in future work, the current submission already makes a meaningful contribution that should be of interest to the ICML community.